# Environmental temperatures shape thermal physiology as well as diversification and genome-wide substitution rates in lizards

Joan Garcia-Porta et al.[#]

Climatic conditions changing over time and space shape the evolution of organisms at multiple levels, including temperate lizards in the family Lacertidae. Here we reconstruct a dated phylogenetic tree of 262 lacertid species based on a supermatrix relying on novel phylogenomic datasets and fossil calibrations. Diversification of lacertids was accompanied by an increasing disparity among occupied bioclimatic niches, especially in the last 10 Ma, during a period of progressive global cooling. Temperate species also underwent a genome-wide slowdown in molecular substitution rates compared to tropical and desert-adapted lacertids. Evaporative water loss and preferred temperature are correlated with bioclimatic parameters, indicating physiological adaptations to climate. Tropical, but also some populations of cool-adapted species experience maximum temperatures close to their preferred temperatures. We hypothesize these species-specific physiological preferences may constitute a handicap to prevail under rapid global warming, and contribute to explaining local lizard extinctions in cool and humid climates.

The intricate relationships between biodiversity and climate have intrigued scientists since the pioneering work of Alexander von Humboldt, over 200 years ago[1]. Climatic conditions, and their change over time and space, influence organisms at multiple levels[2,3]. For instance, numerous traits originated through adaptation to climate[4], present-day biotas have been shaped by paleoclimate[5], and salient climate-related macroecological patterns such as the tropical peak of species richness[6] apply to organisms across the entire tree of life.

Squamates (lizards, amphisbaenians, snakes), are no exception to this rule[7], as is evident from their spectacular diversity of form and function in tropical rainforests and overall rareness in cooler parts of the globe. Species richness in textbook examples of adaptive radiations such as *Anolis* lizards peaks in the tropics[8] and overall, far fewer squamates inhabit the cool areas of the globe[9]. Yet, lizards are different from other vertebrates in that their species richness is also high in arid bioclimatic zones of Australia, Africa, and central Asia[9]. Lizards are typically seen as heliotherms (i.e., gaining energy from controlled sun exposure), and this active thermoregulation might have favored the evolution of a number of species-rich lizard clades in seasonal and temperate biomes.

Many species of lizards worldwide are facing decline, as a probable consequence of a global temperature increase combined with changes in precipitation patterns, habitat loss and fragmentation[10]. Theory predicts that declines should be especially acute in (i) forest-dwelling tropical lizards which often live close to the upper limit of their physiological tolerances and are thermoconformers with a low potential for behavioral thermoregulation, unable to compensate for rising temperatures[10,11]; and (ii) montane microendemics that may be driven to extinction when upslope range shifts become impossible and competitor species move to higher elevations, or when water loss rates and thus physiological stress increase[10,12]. Understanding physiological constraints under which a species operates[13], and the paleoclimatic history under which these constraints evolved, is crucial to improve our ability to predict its response to future climatic change.

A lizard group particularly suitable for integrative research on climate adaptation, combining paleorecords, genomics, physiology and mechanistic models, is the Old World family Lacertidae. Lacertids are the most diverse and ubiquitous squamates in the Western Palearctic[14] containing around 340 species of rather conserved morphology. As predominant lizards in Europe, they have become a well-studied model group in hundreds of physiological, ecological and evolutionary studies, for instance yielding fundamental insights into evolutionary adaptation in diet, morphology, and metabolism[15,16]. Lacertids inhabit diverse habitats, from rainforests to deserts, at sea level and on high mountains. Some species occur in extremely hot and xeric environments in the Namib Desert[16] while one species (*Zootoca vivipara*) has the largest and northernmost distribution of all terrestrial squamates, reaching subarctic regions[17,18]. Because numerous lacertid genera are predominantly distributed in temperate zones that have experienced dramatic climatic changes during the Cenozoic, it can be expected that changing climatic conditions have had profound impacts on their evolutionary history.

In this study we aim to understand if and how the extraordinary diversity of past and present climatic environments experienced by the Lacertidae has shaped their species diversity, physiology, and molecular evolution. We assemble a vast set of novel phylogenomic, physiological and distributional data, and integrate these to provide a wider picture on the effects of climate and climate change on these lizards, and on biodiversity in general.

We set a first goal to achieve a reliable understanding of the evolutionary relationships of lacertids, as a baseline for all further macroevolutionary analyses, using phylogenomic approaches. Lacertids are phylogenetically divided into two major clades, the Gallotiinae and the Lacertinae, and the latter are divided into two clades: the tribe Eremiadini of mainly African-Near Eastern distribution, inhabiting warmer climates, and the tribe Lacertini of mainly European and Asian distribution in cooler climates. Despite enormous progress in lacertid systematics since the pioneering work of Arnold[19] and colleagues, the phylogenetic relationships within Lacertidae are still incompletely resolved[14,18,20]. Very short phylogenetic branches at the base of the Lacertini suggest rapid diversification, jeopardizing phylogenetic inference[20,21]. Lacertid divergence times are controversial as well; for instance, estimates of the Lacertini crown age-range between 15 and 47 Ma[14,18,20,22].

Secondly, we analyze how the contemporary climatic conditions experienced by lacertids are related to their physiological and genomic adaptations. These lizards in general are excellent thermoregulators and thus are expected to compensate behaviorally for suboptimal environmental temperatures[23–25]; they also have been observed to adapt quickly to novel temperature regimes[26]. A study of *Zootoca vivipara* at its northernmost margin found that its preferred temperature was not available in the habitat during much of the day, so thermoregulation (in this case heliothermy) would have been selectively advantageous over thermoconformity despite its activity cost[17,27]. We expect that such heliotherms will be most successful in seasonal environments and that this is reflected in their species richness patterns. Considering records of variation of thermal tolerances and preferences among lacertid species[28], we also predict that their preferred and field active body temperatures, across species, will most strongly be correlated with environmental solar radiation. At the molecular level, change can be expected to accelerate with increasing temperature, and we hypothesize such increased substitution rates would affect the entire genome.

At the macroevolutionary level, we expect that paleoclimate acts as a strong modulator of the adaptive evolution and diversification of lacertids. Given the physiological constraints limiting high-temperature tolerances of animals, the process of adaptation to cold rather than heat would be expected to play a major role[13]. We therefore predict a prevalent origin of thermal adaptations during episodes of global cooling. We also predict that the climatic changes in the Cenozoic will have had a great impact on diversification rates—in particular, episodes of warming should have led to range fragmentation of cool-adapted species and thus have increased diversification rates via allopatric speciation.

Our analysis finds that cold-adapted species have on average smaller distribution ranges, that physiological traits and molecular substitution rates of lacertid species are closely linked to the climatic conditions they experience, and places the origin of these thermal niche adaptations in a period of progressive global cooling. The extinction risk of tropical lizards in the face of anthropogenic climate change has been emphasized previously, and the pervasive physiological adaptations to cold and often small distribution ranges of many lacertids identified herein cast doubt on the ability of these cold-adapted lacertids to persist in increasingly warmer and drier environments.

## Results

**Phylogenomic relationships of lacertid lizards.** The phylogenomic analyses reconstructed the evolutionary relationships among the main groups of lacertid lizards with high support. Concordant results were obtained from analyses of two largely independent, newly acquired datasets (Fig. 1): (i) 6269

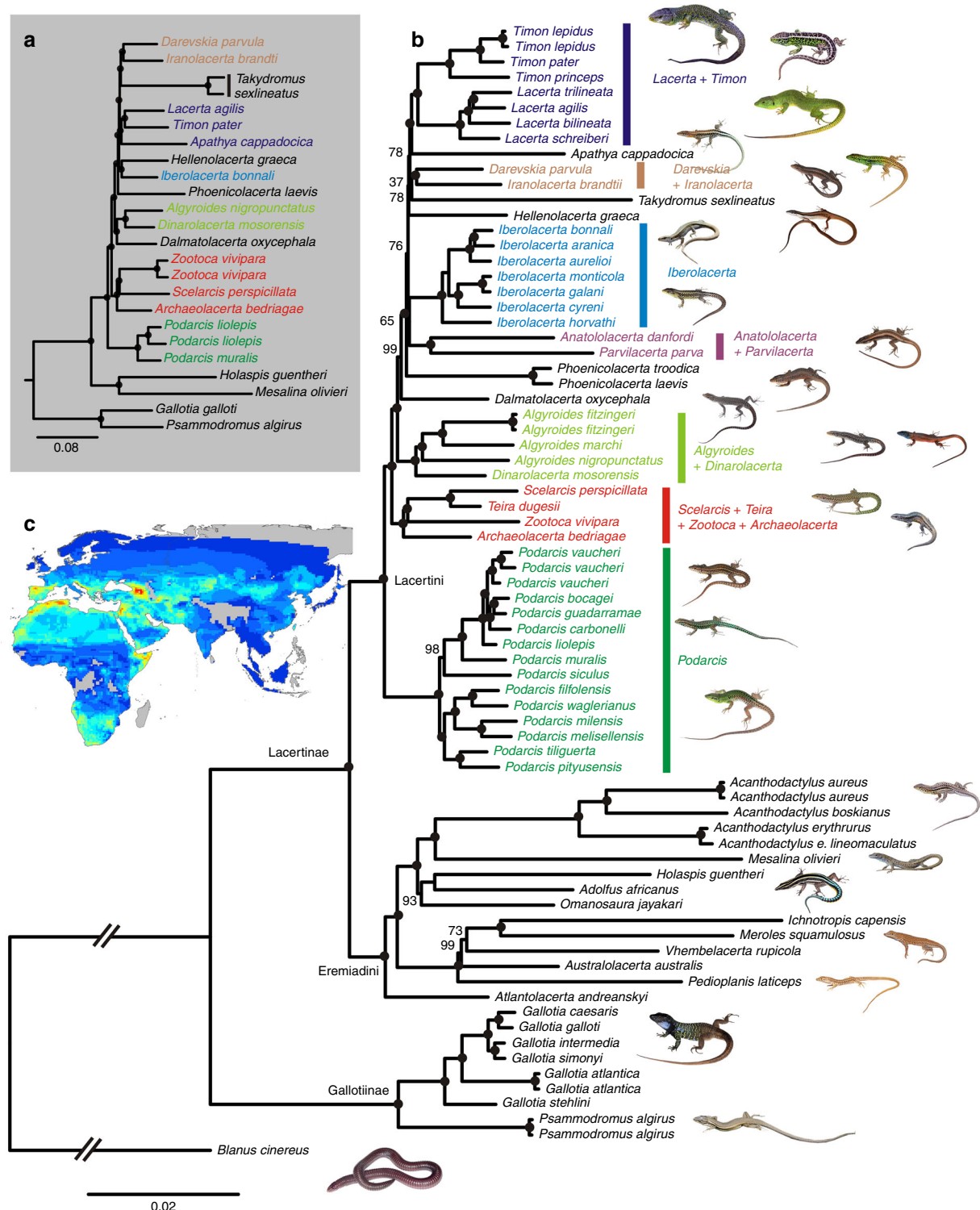

**Fig. 1** Phylogenomic reconstruction of lacertid lizard phylogeny. Maximum Likelihood trees are based on **a** 6269 protein-coding nuclear loci (11,087,328 nucleotide positions) obtained by RNAseq and **b** 324 anonymous nuclear loci (558,418 nt) from anchored hybrid enrichment sequencing. Black dots indicate 100% support from ultrafast bootstrap with lower values provided at relevant nodes. Main clades recovered within Lacertini are indicated with colored bars. **c** Species richness of lacertid species compiled from the Global Assessment of Reptile Distributions[9]; warmer colors represent higher values

protein-coding nuclear loci for 21 lacertid species obtained by adding novel RNA sequencing (RNAseq) data to a previously established set of vertebrate loci[29], and (ii) 324 anonymous nuclear loci for 65 species obtained by anchored hybrid enrichment sequencing (AHE)[30]. Importantly, partitioned maximum likelihood (ML) concatenation and coalescent species tree

analyses, performed separately on each dataset, recovered largely congruent tree topologies (Fig. 1, Supplementary Fig. 2S). All nodes in the RNAseq-ML tree received full bootstrap support, and the same was true for most nodes in the AHE-ML tree. Gene jackknifing proportions (GJP)[29,31], which provide a more stringent test for monophyly than non-parametric bootstrapping,

delivered high support for most nodes in the trees (Supplementary Figs. 6 and 7).

The phylogenomic analyses unambiguously recovered the monophyly of all included genera, the subfamilies Gallotiinae and Lacertinae, and the tribes Lacertini and Eremiadini. Furthermore, we also confirm a series of clades within the Lacertini with strong support, namely those containing (i) the large-sized and green-colored *Lacerta* and *Timon*[14,20,32], (ii) the genera *Darevskia* and *Iranolacerta* distributed in the highlands of Iran, Asia Minor and the Caucasus[20], (iii) *Algyroides* and *Dinarolacerta*[20,21], (iv) the morphologically and ecologically disparate monotypic genera *Archaeolacerta* (a Tyrrhenian rock-dwelling lizard), *Scelarcis* (a North African rock-dwelling lizard), *Teira* (endemic to the volcanic Madeira archipelago in the Atlantic Ocean), and *Zootoca* (the most widespread of all lizards, associated with mesic terrestrial habitats[20]). The latter two clades were also supported by high GJP values, whereas the *Darevskia* + *Iranolacerta* clade received only moderate GJP support (Supplementary Figs. 6 and 7).

Other clades in the tree are unprecedented or have been controversial among previous studies. Most importantly, all phylogenomic analyses supported the species-rich Mediterranean genus *Podarcis* as sister group to all other Lacertini, and the North African montane endemic *Atlantolacerta* as sister group to all other Eremiadini, in both cases also validated by high GJP values. The placement of (i) *Apathya* sister to the *Lacerta*+*Timon* clade, (ii) *Takydromus* sister to *Darevskia*+*Iranolacerta*, (iii) *Dalmatolacerta* sister to *Algyroides*+*Dinarolacerta*, and (iv) *Hellenolacerta* sister to *Iberolacerta*, received comparatively weaker support from GJP, despite high bootstrap values and congruence of summary coalescent and concatenation analyses for all except the last of these clades (Supplementary Figs. 6 and 7). Gene jackknifing thus indicated these clades were not strongly supported by the data. However, their recovery by both AHE and RNAseq analysis, and partly by both concatenated and summary coalescent approaches, suggests they might correctly reflect evolutionary relationships.

To increase species-level representation, we built a large-scale lacertid phylogenetic tree by combining the RNAseq and AHE datasets with DNA sequences of four mitochondrial and one nuclear loci traditionally used in lacertid phylogenetics, selected to maximize taxon coverage. The new dataset, totaling 11,175,421 aligned nucleotide positions (6598 loci) for 262 species (see Supplementary Table 10 for further alignment details and amount of missing data) was analyzed by partitioned ML. Based on a newly compiled set of 89 morphological characters that were in part obtained via newly generated CT-scans, we estimated the phylogenetic position of seven fossil lacertid taxa and subsequently time-calibrated this 262-species tree (Supplementary Figs. 10 and 11) using a penalized likelihood approach. Crown ages were recovered for Lacertidae in the Paleocene at 86.6 Ma, for Gallotiinae at 35.0 Ma, for Lacertinae at 61.3 Ma, and for Eremiadini and Lacertini at 57.1 and 37.6 Ma.

**Dynamics of climatic and physiological evolution**. To test whether the key physiological traits of extant Lacertidae reflect adaptation to the climatic conditions they experience within their range, bioclimatic variables[33] for all 262 species in the timetree were extracted from a newly compiled, georeferenced set of 39,414 lacertid occurrence points. We determined whether the spatial arrangement of current bioclimatic niches predicts lacertid species richness, using data from the Global Assessment of Reptile Distributions project[34] in mixed Spatial Auto-Regressive (SAR) models. We found species richness to be low in aseasonal, equatorial environments (Supplementary Fig. 1), positively

associated with solar radiation and negatively with radiation seasonality (Supplementary Table 3). Lacertini richness peaked along the southern margin of temperate regions of Europe, northern Africa and Asia whereas most species of Eremiadini occurred in rather arid regions of northern and southern Africa and Asia (Supplementary Fig. 1).

Preferred body temperature ($T_{pref}$) and instant evaporative water loss (IWL) were independently assessed for 61 and 51 lacertid species, respectively. Median values per species were compiled from experiments with 792 and 626 individuals, respectively, almost exclusively males during their reproductive season. Phylogenetic mapping and ancestral reconstruction of $T_{pref}$ and IWL showed that, in general, species in Eremiadini have higher $T_{pref}$ and lower IWL than Lacertini (Supplementary Fig. 12). The Eremiadini ancestor likely inhabited warmer regions compared to Lacertini (2393 vs. 1374 h yearly hours >30 °C; Figs. 2a and 3a). Several Eremiadini subclades independently colonized either warm or temperate climates in different geographic regions (e.g., the North African/Asian *Acanthodactylus* adapted to desert environments independently from southern African genera), whereas in the Lacertini, only some *Takydromus* colonized warm (tropical) areas. Not surprisingly, almost all of the bioclimatic variables were characterized by a strong phylogenetic signal in the Lacertidae, as well as in separate analyses of Lacertini and Eremiadini, suggesting that closely related taxa inhabit similar bioclimatic niches (Supplementary Table 4). We also found a comparatively low but statistically significant phylogenetic signal for $T_{pref}$ (Blomberg $K = 0.46$, $P = 0.009$) in Lacertidae. $T_{pref}$ also had a significant phylogenetic signal in Eremiadini ($K = 0.87$, $P = 0.033$) but not in Lacertini ($K = 0.43$, $P = 0.213$). No significant phylogenetic signal was found for IWL in Lacertidae ($K = 0.26$, $P = 0.13$) and Lacertini ($K = 0.32$, $P = 0.492$), while this trait could not be tested in Eremiadini due to the scarcity of available data.

We tested whether physiological traits of lacertids are related to the current bioclimatic properties of their ranges, measured as median values of bioclimatic variables for all occurrence records of a species. Informed by current knowledge on activity temperatures of lacertids (Supplementary Methods), global estimates of microclimate[35] were used to assemble new geospatial layers reflecting (i) the number of yearly hours >30 °C and (ii) the number of yearly hours with a solar radiation >100 W/m$^2$ and a temperature >4 °C. Initial response screening (Supplementary Table 5) revealed that a species' $T_{pref}$ is correlated with various bioclimatic variables characterizing its current distribution range, whereas only weak bioclimatic associations were found with IWL. For $T_{pref}$, our biologically informed variable of yearly hours >30 °C was among the variables with strongest effect sizes and was therefore used for further analysis. In phylogenetic regressions across Lacertidae (Supplementary Table 6), this variable was positively related to $T_{pref}$ ($P < 0.0001$; significant also after Bonferroni correction) and negatively related to IWL ($P = 0.021$). In separate analyses these trends were found in both Lacertini and Eremiadini, but only the relation of IWL with yearly hours >30 °C in Eremiadini was statistically significant (phylogenetic regression; $P = 0.0335$). Body temperatures ($T_b$) of active lizards from a published compilation[36] complemented by our data (Supplementary Table 13) were correlated with $T_{pref}$ (phylogenetic regression; $P < 0.0001$), and IWL with field $T_b$ (phylogenetic regression; $P = 0.0058$) but, possibly due to missing data, not with $T_{pref}$ (phylogenetic regression; $P = 0.2089$).

Multiple phylogenetic regressions were applied against a set of 10 bioclimatic variables selected for least autocorrelation to understand their interaction in predicting physiological traits of lacertids. The analyses (Supplementary Table 7) confirmed that $T_{pref}$ was related to bioclimate and identified solar radiation

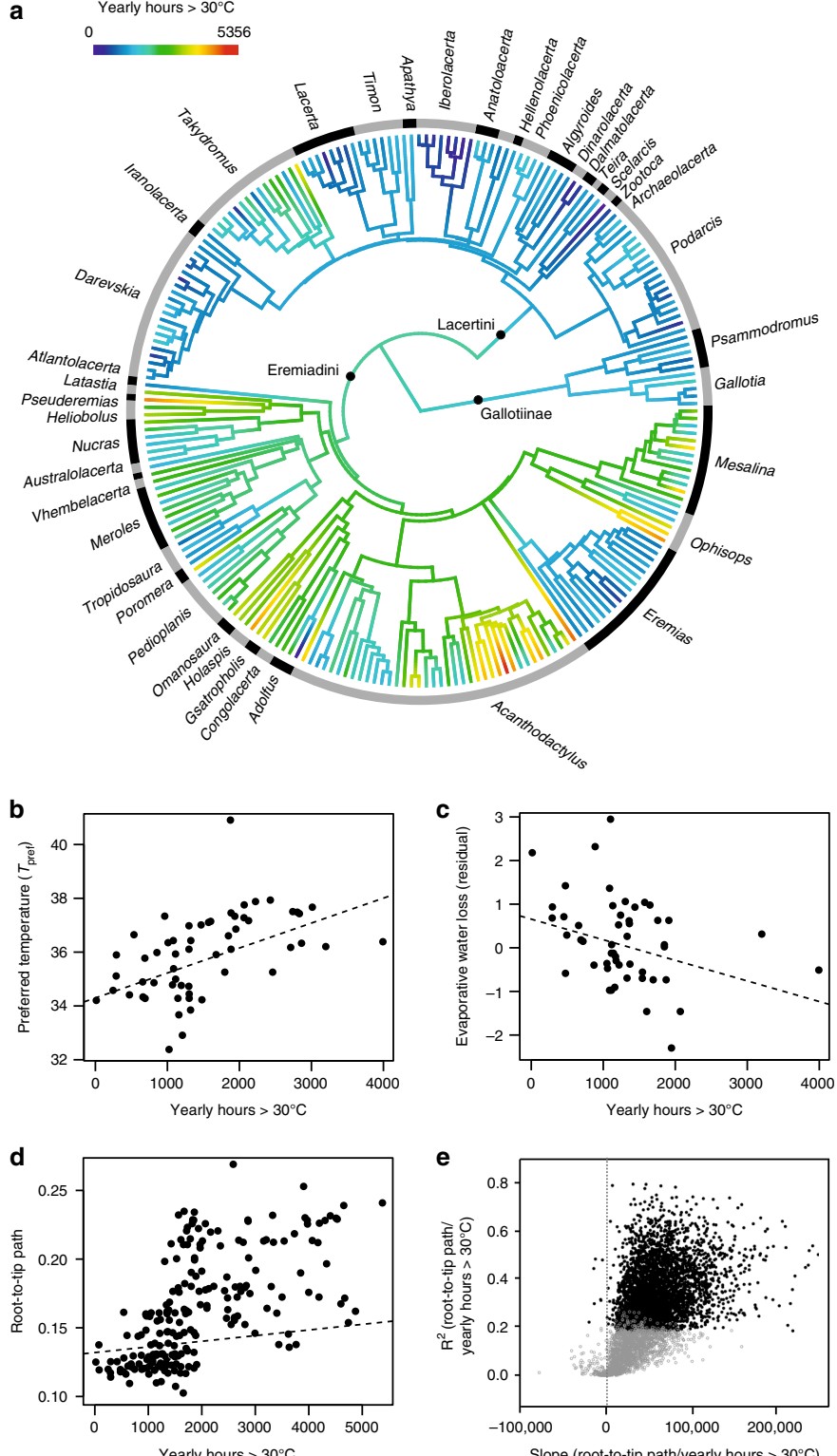

**Fig. 2** Associations between environmental temperature, physiology, and molecular substitution rates of lacertids. **a** Time-calibrated phylogeny derived from partitioned ML analysis of the combined dataset for 262 lacertid species. Colors represent character state reconstruction for yearly hours >30 °C of the species' ranges computed from an occurrence record database. **b–d** Scatterplots among $T_{pref}$ and IWL, and root-to-tip path (a proxy for molecular substitution rate) against a bioclimatic variable (yearly hours >30 °C) derived from distribution range information (all phylogenetic regressions significant; fitted regression lines calculated using a phylogenetic linear model, see main text and test statistics in Supplementary Information). **e** Plot of $R^2$ vs. slope clues for regressions of root-to-tip paths vs. environmental temperature for 5878 gene trees derived from the RNAseq analysis, showing a positive slope for the majority of genes. Black dots are statistically significant correlations at $P < 0.05$. Source data are provided as a Source Data file

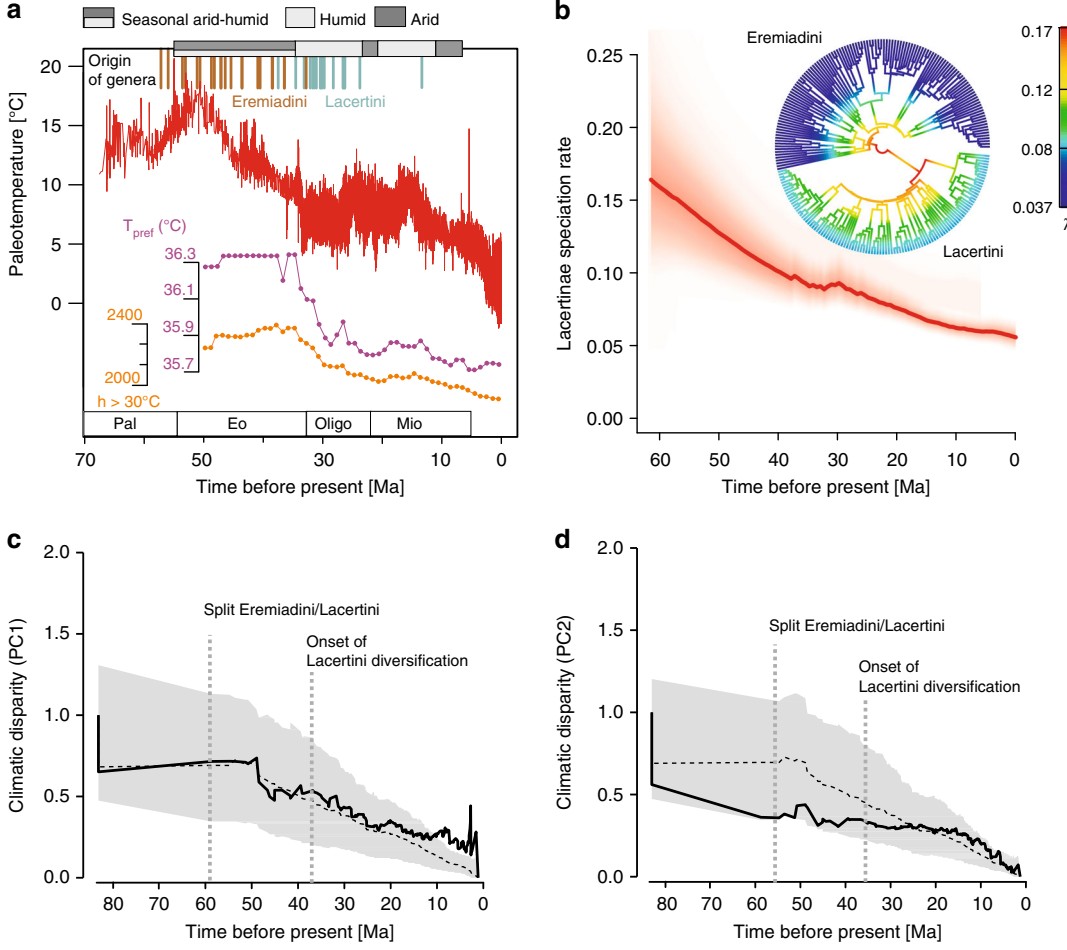

**Fig. 3** Associations between paleoclimate and lacertid diversification. **a** Paleoclimatic reconstructions of temperature (red graph) and humidity (top bars) during diversification of the Lacertinae (Eremiadini+Lacertini); splits among genera within the two tribes shown as brown and blue lines. Orange and purple inset lines show trend of phylogenetically reconstructed $T_{pref}$ and thermal niche (yearly hours >30 °C), averaged over all lineages existing at a certain time as in Fig. 1a and Supplementary S13. **b** BAMM-estimated speciation rates through-time plot and best-fit model of speciation rates plotted on circle tree of Lacertinae. Color density in red shading denotes confidence on diversification rate reconstructions at any point in time. **c**, **d** Plots of climatic disparity through-time (DTT), for the first and second principal components (PC1/PC2) of a phylogenetic principal component analysis (PPCA) of seven bioclimatic variables across lacertids, showing an increase of climatic disparity in PC1 around 45–30 Ma that coincides with the onset of Lacertini diversification. The dashed line indicates the median subclade DTT based on 1000 simulations under a Brownian motion model. The gray shaded area indicates the 95% DTT range for the simulated data

(hours >4 °C and >100 W/m²), and in Lacertini also temperature (hours >30 °C), as significant drivers of $T_{pref}$ variation. IWL was not significantly related to any bioclimatic variable.

Over time, average values of $T_{pref}$ and yearly hours >30 °C consistently decreased starting 35 Ma, documenting the increasing proportion of Lacertini species adapted to cold (Fig. 3a).

Several species of lacertids are distributed over wide ranges whereas others are microendemic to specific mountain ranges, and consequently the breadth of the bioclimatic envelopes they experience within their ranges varies widely. In phylogenetic regressions, range sizes were influenced by body size ($P = 0.0231$), hours >100 W/>4 °C ($P = 0.0496$), and hours >30 °C ($P = 0.0010$) (Supplementary Table 6). Only the latter comparison remained significant after Bonferroni correction, suggesting that species in warmer environments occupy larger distribution ranges.

**Climate-related molecular evolution.** Variation of branch lengths on the lacertid trees (Fig. 1) is suggestive of disparate molecular substitution rates among clades. Phylogenetic regression on the 262-taxon tree (Supplementary Table 6) showed that

root-to-tip paths (an estimate of molecular substitution rates[37]) were strongly positively predicted by yearly hours >30 °C ($P < 0.0001$). This relation was found to be significant also within Lacertini ($P = 0.0035$), and Eremiadini ($P = 0.0048$), and in Lacertidae was maintained in a multiple phylogenetic regression ($P = 0.0310$) which also revealed a weak negative relation with precipitation in the wettest week ($P = 0.0187$). Phylogenetic regression also revealed a positive association of root-to-tip paths with $T_{pref}$ ($P = 0.0124$; Supplementary Table 6). As molecular substitution rates in many animals, including squamates[38], can be influenced by body size (resulting from a generation time effect, as age at maturity increases with body size[38]), we tested if the observed disparity in substitution rates might be influenced by this variable. We found maximum body size not to be a significant predictor of root-to-tip paths (Supplementary Table 6) and the influence of bioclimate (yearly hours >30 °C) was maintained when controlling for body size (Supplementary Table 7). To understand whether this relation is driven by substitution rates of particular genes we first verified that also in the RNAseq tree, species from hotter environments had longer root-to-tip paths (linear regression, $R^2 = 0.46$, $P = 0.0010$; see branch lengths

in Fig. 1a). We then calculated the same regression for each of the gene trees deriving from those 5878 protein-coding nuclear loci in the RNAseq dataset for which the outgroup was available. In 5728 genes the regression slope was positive (linear regression, statistically significant at $P < 0.05$ in 3499 genes), in 150 genes it was negative (significant in only seven genes). In gene trees reconstructed from the respective amino acid sequences, the slope was positive in 4745 (significant in 1387) and negative in 1132 (significant in 26). None of the seven genes with deviant signal at the nucleotide level was under selection (Codon-based $Z$ test of selection of overall averages; non-neutrality rejected at $P < 0.0001$), and visual inspection of the alignments of these genes did not reveal any amino acid substitutions common to unrelated species from cold or hot environments, respectively.

**Dynamics of climatic niche disparity and diversification**. We inferred the disparity of climatic niches associated with lacertid evolution represented by the first two factors of a phylogenetic principal component analysis (accounting for nearly 50% of the total variance; see Supplementary Table 9 for factor loadings). These were derived from the ten least-correlated bioclimatic variables from the lizards' current ranges, with high loadings of temperature and radiation on the first factor (PC1) and precipitation on the second (PC2). The disparity plots (Fig. 3c, d) suggested that disparity of PC1 started exceeding the median of the Brownian Motion (BM) simulations just before the onset of Lacertini diversification about 40 Ma. In both PC1 and PC2 the increase in climatic disparity became more obvious after 20 Ma, coinciding with a period of intense cooling (Fig. 3a); in PC1 the disparity values exceeded those of the BM simulation confidence intervals in more recent times <10 Ma. Both for PC1 and PC2 we found absolute value of standardized independent contrasts were negatively correlated with node ages ($P = 0.02$ for both PCs; Supplementary Fig. 13).

Analyses of diversification dynamics[39] supported models with speciation rate decreasing through time with best-fit for all three clades tested (Lacertinae, Lacertini, and Eremiadini). However, we found differences between tribes regarding the covariation of these decreases in speciation with the paleotemperature across the Cenozoic. In Eremiadini, the best-supported model implies a strong covariation with temperature, with constant extinction through time, while in the younger Lacertini the best-supported model specified that both speciation and extinction rates decreased towards the present, with low support for all models that specified covariation between speciation/extinction rates with temperature (Supplementary Table 8). A complementary analysis using Bayesian Analysis of Macroevolutionary Mixtures (BAMM)[40] despite large uncertainty in the deep past suggested that overall net diversification rate has declined since around 50 Ma (i.e., since the origin of Eremiadini). However, this analysis also suggested an intermittent episode of sharp increase in diversification rate from 30 to 25 Ma representing the early burst of Lacertini diversification (Fig. 3b, Supplementary S13) and coinciding with a mid-Oligocene temperature increase (Fig. 3a). Assuming allopatric speciation in lacertids (Supplementary Table 16), this agrees with the hypothesis of range fragmentation and thus diversification of cool-adapted species by past global warming.

**Thermoregulation and thermal safety margins**. As $T_{pref}$ converges on ambient daily maximum air temperature ($T_{max}$), a typical heliothermic lizard can assume a more thermoconforming behavior. In lacertids, the difference between $T_{pref}$ and $T_{max}$ averaged across months, indicates larger differences between $T_{pref}$ and $T_{max}$ (typical of heliotherms) in species outside the equatorial tropics (Supplementary Fig. 17). Instead, for many occurrence records at absolute latitudes <30, $T_{pref}$ falls very close to $T_{max}$, suggesting that tropical lacertids need to spend less time thermoregulating and have narrower thermal safety margins when confronted with global warming. Populations of cool-adapted species differ widely in this thermal safety margin metric, and for one of these, *Zootoca vivipara*, recently extirpated populations had distinctly lower values than other, extant populations. For this species, extinctions have been recorded at sites where the thermal safety margin is less than ~12 °C (Supplementary Fig. 17).

**Discussion**

This study provides a reliable, well-supported hypothesis of the phylogenetic relationships among the Lacertidae, and thereby contributes a critically missing baseline for a plethora of ongoing research projects in this intensively studied Palearctic group of lizards. Despite lacertids (and in particular Lacertini) being a clear example of rapid diversification[21], the congruence of multiple phylogenomic approaches in our study suggests the estimated phylogeny is accurate and robust. In particular, the concordance between the topologies obtained by concatenation and summary coalescent approaches—not a self-evident pattern[41]—lends substantial confidence to our phylogenomic resolution of the deep relationships among lacertids. We found strong support for clades that contain morphologically, biogeographically and ecologically distinct taxa[20], of which perhaps most striking is the association of the ecologically and morphologically disparate genera *Archaeolacerta*, *Teira*, *Scelarcis*, and *Zootoca*. Overall, our results confirm the efficacy of phylogenomics to infer systematic relationships that have long evaded resolution.

Thermoconformity is predicted to have low selective benefits under cool temperature regimes[17,27] and it is reasonable to explain the success of lacertids in such environments with their obvious thermoregulatory, heliothermic basking behavior[28], although previous work has characterized some lacertid species as partial thermoconformers[42]. To facilitate a more efficient thermoregulation in cooler environments, an evolutionary trend towards lower $T_{pref}$ can be expected.

In agreement with these hypotheses, our spatial analysis found lacertid species richness positively associated with solar radiation, and key physiological traits of extant Lacertidae correlated with the climatic conditions they experience, indicating they have physiologically adapted to these climatic conditions. The phylogenetic signal of $T_{pref}$ across Lacertidae, in agreement with previous studies across lizards[43], supports that $T_{pref}$ adaptations are characteristic of major clades that diverged early in lacertid evolution.

Our lacertid analysis provides evidence for a correlation of physiology and climate across an entire squamate clade. Although this pattern in itself is in agreement with our hypotheses and thus not surprising, the consistency by which environmental temperature was correlated with other traits in our family-wide analysis is striking. For instance, environmental temperature was correlated to experimental $T_{pref}$, a biologically relevant trait influencing the temperatures under which these lizards are active in the field, as confirmed by its close association to independently measured field $T_b$. Also, the spatial extent of a species' distribution area was strongly related to environmental temperature, again suggesting this variable profoundly influences lacertid biodiversity patterns. On the contrary, we found only a weak correlation between range size and body size among lacertid species although such range size-body size relationships prevail in other organisms[44,45], including reptiles[46]. Environmental temperature has furthermore affected molecular evolution of lacertids. Substitution rates were significantly slower in lineages

evolving in cool climates (Fig. 2d, e). This exceptionally clear pattern conforms to the metabolic theory of ecology[47], and was not observed in a large-scale study across squamates[47,48]. In a previous phylotranscriptomic study of vertebrates[29], the included lacertid (*Podarcis liolepis*) had a relatively short branch compared to other squamates, thus suggesting its substitution rates are slower than the squamate average. In our study, *Podarcis* likewise was among the taxa with relatively short root-to-tip paths. We hypothesize that this condition is derived, and indicative of a genome-wide slowdown of substitution rates in those lacertids that adapted to cooler conditions, rather than an acceleration in heat-adapted species, which contrasts with previous findings in a cold-adapted agamid lizard that has accelerated substitution rates in several genes[49].

Taking advantage of our phylotranscriptomic dataset, we verified that the positive correlation between environmental temperature and substitution rates affects the vast majority of coding genes across the genome, with only very few genes showing an inverse relationship (i.e., accelerated rates in cool environments; Fig. 2e). This pattern most likely can be explained by an influence of metabolic rate on mutation rate[47], and we did not find any indication that deviations from the overall pattern were caused by selection. As in other organisms[50], the genome-wide differences in substitution rates were also found at the amino acid level and therefore might have adaptive consequences. Decelerated mutation rates could reduce the speed by which genetic variation arises and thus influence standing genetic variation, which probably determines in great part the ability for fast adaptation[10,51]. It therefore would be tempting to speculate that the detected slow substitution rates in temperate lacertids could be associated with a reduced adaptive potential to environmental change, but fast climate adaptation has been observed in temperate lacertids experiencing drastically changed climatic conditions[26,52]. The possible macroevolutionary consequences of the observed slowdown in substitution rates of cold-adapted species, and the role of variation in protein-coding vs. non-coding loci warrant further study.

In comparison with the very clear temperature-related patterns encountered in our dataset, the signal of humidity regimes appeared weaker. Precipitation-related variables usually played a secondary role predicting physiological and genomic traits or range sizes, and we found that IWL was phylogenetically less conserved than $T_{pref}$ (as indicated by the absence of a significant phylogenetic signal in IWL), despite IWL being correlated to climate. This may indicate a stronger selection pressure on and faster adaptation of this key physiological trait, which might be less efficiently buffered by behavior than thermal preferences. This hypothesis would predict that intraspecific local adaptation of widespread lacertids should be more strongly reflected in variation of IWL than $T_{pref}$, which could be easily tested in future meta-analyses. In addition, it will be paramount to obtain IWL data for more arid-adapted desert species in the Eremiadini, as we assume that sampling gaps might have obscured possible signals in our data.

Lacertids, and in particular the Lacertini, stand out among major squamate clades in having diversified largely outside equatorial regions and ranging deep into the temperate zone. Adaptation to cool environments has been predicted to be a primary driver of lizard evolution[13] and will have repercussions on their performance in future climates. Our phylogenomic tree confirms that multiple lacertid clades have independently conquered cold environments. These clades include the Lacertini genus *Zootoca*, whose range extends into the Subarctic, as well as the montane genera *Iberolacerta*, *Darevskia*, and *Dinarolacerta*. Within the Eremiadini, they include *Atlantolacerta* and *Eremias*. These and other lineages have adapted to climates with 0–700 h

above 30 °C per year, and most of them diverged from other genera after the Eocene-Oligocene boundary. This epoch coincided with a large-scale faunal turnover and aridification in some parts of the world[53,54] but not in others[55,56]. In Europe, this period probably was relatively cool, humid and stable. To understand adaptation processes in the sub-Saharan Eremiadini clades that experienced this aridification, we would need to fill the gaps of ecophysiology data on taxa that secondarily entered cold, mesic environments (i.e., *Tropidosaura*, *Australolacerta*, and *Vhembelacerta*).

We hypothesize that the current bioclimatic niches of most Lacertini, and thus probably their current physiology represented by a relatively low $T_{pref}$, originated in the Oligocene and was further shaped in the continued cooling at the end of the Miocene. During this period, we detected an increasing disparity of bioclimatic niche diversification (Fig. 3c, d) combined with decreased average values of $T_{pref}$ and thermal niche across lacertids (Fig. 3a). This scenario of pervasive cold adaptation in Lacertini is consistent with the fossil record in Europe, where several lizard groups disappeared while lacertids persisted throughout the Oligocene-Miocene.

Our analysis reveals a strong association of present environmental temperature to physiological traits ($T_{pref}$, IWL), molecular substitution rates, range sizes, and diversification of lacertids. The current trend of rapidly rising temperatures will likely alter the adaptations of preferred to environmental temperatures that have been shaped through millions of years. The capacity of species to buffer these new conditions through behavioral responses remains unknown.

It is possible that because of their small thermal safety margins, many tropical lacertids may be perilously close to at least local extinction events under future climate scenarios where $T_{max}$ would exceed $T_{pref}$. This suggests a need for conservation assessment and thermophysiological work especially on the largely understudied lacertid taxa occurring in aseasonal tropical environments. Yet, current declines among lacertids seem to especially affect specialists inhabiting montane, and/or cool and moist environments[10,57] such as *Z. vivipara* (Supplementary Fig. 17b), whereas species with very high $T_{pref}$ such as *Pedioplanis husabensis* so far appear to remain unaffected by contemporary climate change[25]. As one hypothetical explanation, species in cool environments also are characterized by high IWL and thus may be forced to thermoregulate sub-optimally or become inactive when conditions become too dry. Despite their low $T_{pref}$, they will rarely be able to behave as full thermoconformers in temperate environments. This could explain declines detected in species such as *Algyroides marchi* and *Podarcis carbonelli*[58]. In the viviparous populations of *Z. vivipara*, the species with highest IWL in our dataset, we speculate that this handicap could be exacerbated in gravid females as these require optimal thermoregulation to ensure embryonal development for a longer period of time than oviparous species. Many cold-adapted species furthermore are restricted to very small ranges with an associated increased extinction risk, as revealed by the overall correlation of range size to temperature we observed.

Hence, we hypothesize that not only tropical lizards, but also some lacertids from mesic and temperate zones might be "toast" in the face of climate change[59], although they are able to cope with substantial annual and diurnal variation in temperature and humidity, and may not operate near their critical thermal safety margins just yet. We recommend a conservation and research focus on two groups of lacertids: (i) those occurring in the tropics, and (ii) montane microendemics whose extirpation risk may be particularly strong through the combined effects of rising temperature, rising aridity, and increased competition by other species better adapted to the novel thermal and hydric conditions.

## Methods

**Phylogenomics.** We assembled two main datasets (AHE and RNAseq) to resolve lacertid phylogeny and combined these with five gene fragments commonly used in lacertid phylogenetics. The amphisbaenian *Blanus*, a representative of the sister group of lacertids[60] was used as outgroup. AHE sequencing followed established methods for squamates[30,61]. For RNAseq, RNA was trizol-extracted from pooled samples of different organs (skin, muscle, and liver of single individuals per species). Libraries were prepared following the Illumina TruSeq mRNA protocol, and sequenced on an Illumina NextSeq platform. After read pre-processing, transcriptomes were assembled de novo with Trinity v. 2.1.0 (ref. [62]). Ortholog sequences were selected, translated to amino acids and aligned to a previously compiled set of markers across vertebrates[29] using the software Forty-Two (https://bitbucket.org/dbaurain/42/). Sequences from non-vertebrate sources, cross-contaminations, misaligned and possibly paralogous sequences were removed[29] (Supplementary Methods) and nucleotide sequences for the retained amino acid alignments recovered from the original assemblies. A third dataset was compiled with sequences from previous studies for the nuclear proto-oncogene mos gene (*c-mos*) and the mitochondrial genes for 12S and 16S rRNA, Cytochrome *b*, and NADH-dehydrogenase subunit 4. For those species included in the phylogenomic dataset we assembled full mitochondrial genomes, and extracted the four mitochondrial target genes.

**Phylogenetic analyses.** Maximum likelihood trees were inferred separately for the AHE and RNAseq datasets using IQTREE v. 1.5.4 (ref. [63]). Best-fitting partitioning schemes and substitution models were selected based on the Akaike Information Criterion (AIC) using the heuristic algorithms implemented in IQTREE and branch support was assessed by 1000 replicates of ultrafast bootstrapping. For computational feasibility, the combined dataset was analyzed using the best-fit partitions previously selected for the three datasets and best-fit models were selected among JC, HKY, or GTR models, with or without gamma parameter, and assuming edge-proportional partitions ("-spp" option). For the combined dataset containing AHE, RNAseq and the five additional gene fragments, the ML tree was estimated as above except for the constraint to satisfy the topologies recovered by AHE and RNAseq ML trees. For the AHE and RNAseq datasets, we estimated summary coalescent species trees under the multi-species coalescent with ASTRAL II[64] with node support measured by multilocus bootstrapping with gene and site resampling[64]. Single-locus trees were first estimated under the best-fitting model and 1000 replicates of ultrafast bootstrapping.

Gene jackknife analyses[31] were performed to more stringently test tree topologies and understand the amount of data required to recover each bipartition with confidence (>75 GJP). We generated 100 alignment replicates by randomly sampling loci, without replacement, to ca. 10K, 100K, 1000K, and 10,000K nucleotide positions (RNAseq), or 5K, 10K, 50K, 100K, and 500K nucleotide positions (AHE). Each replicate was analyzed by locus-wise partitioned ML and gene jackknife proportions were estimated as the number of times a given bipartition is recovered per replicate length.

**Phylogenetic placement of fossils and molecular dating.** To generate a lacertid timetree based on phylogenetically tested fossil calibrations, we compiled a morphological dataset of 89 characters, based on published data[14,19] and novel microCT scans for 250 specimens of 82 lacertid species of 36 genera. Added to this were data from nine fossil taxa selected according to fossil completeness and stratigraphic reliability (see Supplementary Methods). We pruned our molecular tree for those taxa represented by morphological data and used it as topological constraint in a maximum parsimony analysis of the morphological data in TNT 1.5.3 (ref. [65]), using 100 replicates of traditional and "new technology" tree searches. After excluding two fossils that could not be reliably placed due to their low number of scored characters, we obtained 70 (traditional) and 11 (new technology) equally parsimonious trees (tree length: 1166 steps; Consistency Index: 0.105; Retention Index: 0.484). The consensus topology obtained was used to define the following fossil calibration points (detailed rationale in Supplementary Methods): (i) Lacertidae, 40.4 Ma minimum age (oldest record of *Plesiolacerta lydekkeri*), 150.0 Ma maximum age. (ii) Lacertinae, minimum 33.9 Ma (placement of *Succinilacerta* specimens within Eremiadini), maximum 61.6 Ma (oldest-known, Paleocene lacertid fossils). (iii) Gallotiinae: minimum 28.1 Ma (placement of the fossil clade containing *Dracaenosaurus*, *Pseudeumeces*, and *Janosikia* sister to *Gallotia*), maximum 61.6 Ma; (iv) *Lacerta* s. str.: minimum 4.4. Ma (oldest reliable fossil of the *Lacerta agilis* lineage), maximum 61.6 Ma; (v) *Acanthodactylus erythrurus* – *A. lineomaculatus*[18]: minimum 0.8 Ma (oldest record of *A. erythrurus*), maximum 61.6 Ma. These calibrations were implemented in a penalized likelihood timetree inference, using TreePL 1.0 (ref. [66]) on the full ML phylogram derived from the combined molecular dataset. As any timetree, our hypothesis contains multiple uncertainties but this will not affect many of the downstream analyses (e.g., phylogenetic regressions) because these are independent from the absolute timescale in the tree.

**Ecophysiology and bioclimate.** All sampling and experimental work was performed with the appropriate permits: in Croatia, by the Ministry of Environment and Energy (UP/I-612-07/16-48/11, 517-07-1-1-116-3 from 10 February 2016); in

France (by DREAL Aquitaine 41-2016, DREAL Languedoc Roussillon #2013-274-0002, DREAL Midi-Pyrénées #81-2013-05, DREAL Auvergne #2013/DREAL/259, Prefecture de l'Ariege O.B./01/244); in Germany, of the Landkreis Helmstedt; in Greece (permits AΔA: 73OΣ4653ΠI8-7ΨB and AΔA: ΩΛΔΠ465ΦΘH-Y5E); in Italy, by Regione Sicilia (DPR 357/97); in Morocco, by the High Commissariat for Water and Forest (05/2013 HCEFLCD/DLCDPN/DPRN/CFF and 14/HCEFLCD/DLCDPN/DPRN/CTT); in Namibia, by the Ministry of Environment and Tourism (MET) (permits 1710/2012, 1890/2014, export permits 92349 and 98784); in Portugal, by the ICNF (68I2011ICAPT, 171-180/2012/CAPT, 2704/2013/DRNCN/DGEFF, 2289/2013/DRNCN/DGEFF, 19452/2014/DRNCN/DGEFF, 459I2015ICAPT, 27445/2016/DRNCN/DGEFF) and Parque Natural da Serra da Estrela (00/PNSE); in Serbia, by Ministry of Energy, Development and Environmental Protection, permit No. 353-01-312/2014-08; in Slovenia, by the Slovenian Environment Agency (permits no. 35601–32/2010–6, 35601–47/2011–6, 35601-14/2013-5, and 35601–47/201-6); in Spain, by Diputación General de Aragón (FHF / E-33 issued 23 June 2000), Principado de Asturias, (2000-11834/2000), Comunidad Autónoma de Madrid (10/091126.0/00, 10/069021.9/15, 10/040449.9/13 and 10/091949.9/14), by Xunta de Galicia (#261 issued 11 August 2000, #2791 issued 07 March 2001, EB-041/2017 and 012/2011), by Junta de Castilla-La Mancha (OAEN/SVSlA/avp_10_174_aut, OAEN/SVSlA/avp_10_169_aut, DGAPYB/SB/avp_11_014, DGMEN/SEN/avp_12_163_aut), by Junta de Castilla-León (EP/CYU122/2011, EP/CYL/228/2012, EP163/2000, EP 53/2001, EP 157/2000), by Generalitat de Catalunya (SF/153, SF/154, SF/698), by the region of Murcia (AUT/CAP/UND/44/10, AUTICA PI UND/24/2011, AUT/FA/UND/30/2012), by the Illes Balears region (CAP 55/2011), by the Junta de Andalucia (SGYB/FOA/AFR/2010, SGYB/FOA/AFR/2011), by Cabildo de Tenerife (AFF 160/18), and by Parque Nacional de Ordesa y Monte Perdido (#402 issued 03 May 2000); and in Turkey, by the Ministry of Forestry and Water Affairs, Directorate of Nature Conservation and National Parks (permit number: 2014-51946). Ethics approval was granted by Landesamt für Verbraucherschutz und Lebensmittelsicherheit in Lower Saxony, Germany (Az. 33.19-42502-04-15/1900), ethics committee of the Institute for Biological Research "Siniša Stanković" (05-05/14) in Serbia, and Ege University Animal Experiments Ethics Committee, Turkey (approval number: 2014- 002); accreditation in animal experimentation was issued to M.A. Carretero by Generalitat de Catalunya (Spain).

Evaporative water loss was measured housing lizards individually in plastic boxes and placing these into a large container provided with silica gel to maintain low air humidity at 10–20%, in the dark and at a constant temperature of 20–22 °C[67]. Lizards were weighed once every hour during 6–12 h, and instantaneous evaporative water loss (IWL) calculated as the amount of hourly weight loss, averaged over all measurements per individual, corrected by regression against the lizard's body surface area, estimated as ln[surface area] = 2.36 + (0.69 × ln[mass])[68]. Preferred body temperatures ($T_{pref}$) were estimated as those selected by individuals on a photothermic gradient 20–55 °C[25] created with an incandescent 100 W light bulb. After a short period of acclimation when body temperature rises above room temperature (15 min), lizard body temperature was measured every minute over a period of up to 150 min by thermocouples (details in Supplementary Methods) and $T_{pref}$ was calculated as average over all included measurements. Ecophysiological measurements were obtained for an average of 13 ($T_{pref}$) and 12 (IWL) male individuals per species (detailed sample sizes per species in Supplementary Table 13), often from multiple locations, and median values per species were used for downstream analyses.

We compiled a curated dataset of georeferenced data points for lacertid lizards by combining (i) data from the Global Biodiversity Information Facility (www.gbif.org), (ii) our own published and unpublished datasets, and (iii) literature data. To obtain an approximation of bioclimatic conditions characterizing the overall range of species, we extracted 29 bioclimatic variables (http://www.worldclim.org) at 30 arc seconds of spatial resolution[33]. We calculated yearly hours above 30 °C at 1 cm above the ground, rock surface, and at full sun exposure from the Microclim dataset[35] at a spatial resolution of ~18 × 18 km, as well as yearly hours >4 °C and with solar radiation of >100 W/m². This computation is conceptually similar to the one used to predict the observed extinctions of lizards at global scales[16], differing slightly in *y* and *x*-variable scales.

To analyze the disparity between $T_{pref}$ and the temperatures a species experiences within its range we extracted from Worldclim the monthly maximal and minimal temperatures ($T_{max}$, $T_{min}$), as an average value for each occurrence record for the years 1950–2000. We then excluded months with $T_{min}$ <5 °C, averaged $T_{max}$ across remaining months for each occurrence record, and subtracted this value from the $T_{pref}$ of the respective species.

**Comparative and diversification analyses.** Bioclimatic and physiological variables were mapped onto the lacertid timetree (pruned for species with missing data for each variable), estimating states in internal nodes by ML, assuming a Brownian model of evolution and interpolating them along branches (function contMap in the R package phytools[69]). We explored the relationship between IWL, $T_{pref}$ and bioclimate by means of phylogenetic linear and second order polynomial regression models (PRM) with the R package phyloml[70] selecting a fixed-root Ornstein-Uhlenbeck (OU) model based on AIC scores. Three values (body size, distribution range, and hours >4 °C and >100 W/m²) were log-transformed before analysis. We used Blomberg's *K* statistic[71] to estimate phylogenetic signal. This was done with

the function phylosig in phytools[69], estimating *P*-values by 1000 simulations. To visualize trends of $T_{pref}$ and hours >30 °C over time, we estimated these variables for each branch as the average between the two contacting nodes, and averaged the values for all lineages (branches) existing at a certain time.

To explore the dynamics of species diversification we fitted 15 alternative models of species diversification to the lacertid phylogeny using the R package RPANDA[39]. Of these models six assumed covariation between speciation rates and time, six others assumed covariation between speciation rates and paleotemperature across the Cenozoic and three models assumed constant speciation rates. Across models, extinction was set to zero, either constant or linearly covarying with time or temperature. We subsequently ranked models according to their AIC values. We also independently explored dynamics of diversification using the program BAMM[40] using two independent runs (Supplementary Methods). Temporal dynamics of bioclimatic diversification were visualized by plotting the climatic disparity through-time[72], as well as by calculating the absolute values of standardized independent contrasts of climate values across the phylogeny and regressing it against the associated node ages[73].

**Reporting summary**. Further information on research design is available in the Nature Research Reporting Summary linked to this article.

## Data availability

RNA-Seq data are available from the Sequence Read Archive (SRA Bioproject PRJNA543749), DNA sequences from GenBank (MN015052–MN015359, MN030176–MN030252), and phylogenetic datasets, microCT scans, and physiological experiment raw data in Figshare (DOIs: https://doi.org/10.6084/m9.figshare.8150690.v1/ https://doi.org/10.6084/m9.figshare.8866271.v1/https://doi.org/10.6084/m9.figshare.8863520.v1). The source data underlying Fig. 2b–e and Supplementary Fig. 17a, b and Supplementary Fig. 13 are provided as a Source Data file.

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

## Acknowledgements

We are grateful to numerous students, field assistants and technicians who supported field and laboratory work, and to G. Jones for useful comments on the manuscript. This study was supported by the Deutsche Forschungsgemeinschaft (DFG) to M.V. and J.M. (VE 247/11-1/MU 1760/9-1), and to L.R. in the framework of the "TaxonOmics" priority program (VE 247/16-1 – HO3492/6-1). J.G.-P. and I.I. were supported by Juan de la Cierva fellowships from the Spanish 'Ministerio de Economía y Competitividad' (FJCI-2014-20380 and IJCI-2016-29566), D.S. by the 'Rita Levi Montalcini' program for recruitment of young researchers at the University of L'Aquila, M.A.C. by project NORTE-01-0145-FEDER-000007, B.S. and D.B.M. by the US-National Science Foundation Emerging Frontiers program (EF-1241848), J.C.I. by project ON173025 MESTD RS, T.S., A.F., and A.S. by the Hassan II Academy of Sciences and Technologies (ICGVSA Project), A.Ž. by the Slovenian Research Agency Research Program P1-0255. The computations were in part performed on the Altamira supercomputer at the Institute of Physics of Cantabria (IFCA-CSIC), Spain; the Uppsala Multidisciplinary Center for Advanced Computational Science (UPPMAX) under Project SNIC 2017/7-275; as well as the Zentraler Informations- und Datenverarbeitungsservice of the Tierärztliche Hochschule Hannover (IDS-TiHo). Further, we acknowledge the Viper High Performance Computing facility of the University of Hull and its support team, especially Ahmed Elnawasany, for facilitating computational analyses.

## Author contributions

J.G.-P., I.I., M.K., A.R., S.K., J.L.B., A.M., H.P., A.P.T., and K.C.W.V. performed crucial laboratory work or statistical analyses. F.A., G.A, J.C.-I., I.D.l.R., A.F., P.G., B.G., D.J.H., O.J.-R., U.J., O.J.G., M.K., G.K., S.K., M.L., D.M., M.J.N.H., M.A.O., P.P., L.R., N.R., B.R.C., E.S., D.S., T.S., A.S., A.T.Q., and A.Z. performed experiments and fieldwork. M.V., K.C.W.V., J.M., and B.S. designed the study. A.L., E.M.L, M.A.C., S.C., H.P., B.S., and J.M. contributed to and guided laboratory and statistical analyses. M.V., K.C.W.V., J.G.-P., and I.I. drafted the manuscript, and all authors read and approved the final manuscript.

## Additional information

**Competing interests:** The authors declare no competing interests.

Joan Garcia-Porta[1,32,33], Iker Irisarri [2,33], Martin Kirchner[3], Ariel Rodríguez [4], Sebastian Kirchhof[3], Jason L. Brown[5], Amy MacLeod[3], Alexander P. Turner[6], Faraham Ahmadzadeh[7], Gonzalo Albaladejo[8],

Jelka Crnobrnja-Isailovic[9], Ignacio De la Riva[10], Adnane Fawzi[11], Pedro Galán[12], Bayram Göçmen [13], D. James Harris[14], Octavio Jiménez-Robles [15], Ulrich Joger[16], Olga Jovanović Glavaš[17], Mert Karış[18], Giannina Koziel[19], Sven Künzel[20], Mariana Lyra [21], Donald Miles[22], Manuel Nogales [8], Mehmet Anıl Oğuz[13], Panayiotis Pafilis[23], Loïs Rancilhac[19], Noemí Rodríguez[8], Benza Rodríguez Concepción[8], Eugenia Sanchez[19], Daniele Salvi [14,24], Tahar Slimani[11], Abderrahim S'khifa [11], Ali Turk Qashqaei[7], Anamarija Žagar[25], Alan Lemmon [26], Emily Moriarty Lemmon [27], Miguel Angel Carretero[14], Salvador Carranza[28], Hervé Philippe[29], Barry Sinervo [30], Johannes Müller[3], Miguel Vences [19,33] & Katharina C. Wollenberg Valero [31,33]

[1]CREAF, 08193 Cerdanyola del Vallès, Spain. [2]Department of Organismal Biology, Uppsala University, Norbyvägen 18D, 752 36 Uppsala, Sweden. [3]Museum für Naturkunde, Leibniz Institute for Evolution and Biodiversity Science, Invalidenstr. 43, 10115 Berlin, Germany. [4]Institute of Zoology, Tierärztliche Hochschule Hannover, Bünteweg 17, 30559 Hannover, Germany. [5]Department of Zoology, Southern Illinois University, Carbondale, IL, USA. [6]School of Engineering and Computer Science, University of Hull, Cottingham Road, HU6 7RX Kingston-Upon-Hull, UK. [7]Department of Biodiversity and Ecosystem Management, Environmental Sciences Research Institute, Shahid Beheshti University, G.C, Tehran, Iran. [8]Instituto de Productos Naturales y Agrobiología (IPNA), Consejo Superior de Investigaciones Científicas (CSIC), c/Astrofísico Francisco Sánchez, 38206 Tenerife, Canary Islands, Spain. [9]Department of Biology and Ecology, Faculty of Sciences and Mathematics, University of Niš, Višegradska 33, 18000 Niš, Institute for Biological Research "S. Stanković" University of Belgrade, Despota Stefana 142, Belgrade 11000, Serbia. [10]Department of Biodiversity and Evolutionary Biology, Museo Nacional de Ciencias Naturales, CSIC, C/José Gutiérrez Abascal 2, 28006 Madrid, Spain. [11]Faculty of Sciences, Biodiversity and Ecosystem Dynamics Laboratory, Cadi Ayyad University, Marrakech, Morocco. [12]Departamento de Bioloxía, Facultade de Ciencias, Universidade da Coruña, Grupo de Investigación en Biología Evolutiva (GIBE), 15071 A Coruña, Spain. [13]Zoology Section, Biology Department, Faculty of Science, Ege University, 35100 Bornova, Izmir, Turkey. [14]CIBIO-InBIO, Centro de Investigação em Biodiversidade e Recursos Genéticos, University of Porto, Campus Agrário de Vairão, 4485-661 Vairão, Portugal. [15]Department of Ecology and Evolution, Research School of Biology, The Australian National University, Canberra, ACT, Australia. [16]Staatliches Naturhistorisches Museum, Braunschweig, Germany. [17]Department of Biology, University of Osijek, Cara Hadrijana 8A, Osijek, Croatia. [18]Department of Chemistry and Chemical Process Technologies, Acıgöl Vocational High School of Technical Sciences, Nevşehir Hacı Bektaş Veli University, 50300 Nevşehir, Turkey. [19]Zoological Institute, Braunschweig University of Technology, Mendelssohnstr. 4, 38106 Braunschweig, Germany. [20]Max Planck Institute for Evolutionary Biology, Plön, Germany. [21]Departamento de Zoologia, Instituto de Biociências, UNESP – Universidade Estadual Paulista, Rio Claro, Brazil. [22]Department of Biological Sciences, Ohio University, Athens, OH 45701, USA. [23]Section of Zoology and Marine Biology, Department of Biology, National and Kapodistrian University of Athens, Panepistimioupolis, Ilissia, Athens 157-84, Greece. [24]Department of Health, Life and Environmental Sciences, University of L'Aquila, 67100 Coppito, L'Aquila, Italy. [25]National Institute of Biology NIB, Department of Organisms and Ecosystems Research, Vecna pot 111, 1000 Ljubljana, Slovenia. [26]Department of Scientific Computing, Florida State University, Dirac Science Library, Tallahassee, FL, USA. [27]Department of Biological Science, Florida State University, Tallahassee, FL, USA. [28]Institute of Evolutionary Biology (CSIC-Universitat, Pompeu Fabra), Passeig Marítim de la Barceloneta 37-,49, 08003 Barcelona, Spain. [29]Centre for Biodiversity Theory and Modelling, UMR CNRS 5321, Station of Theoretical and Experimental Ecology, 09200 Moulis, France. [30]Department of Ecology and Evolutionary Biology, and Institute for the Study of the Ecological and Evolutionary Climate Impacts, University of California, 130 McAllister Way, Coastal Biology Building, Santa Cruz, CA 95064, USA. [31]Department of Biological and Marine Sciences, University of Hull, Cottingham Road, HU6 7RX Kingston-Upon-Hull, UK. [32]Present address: Department of Biology, Washington University in Saint Louis, St. Louis, MO 63130, USA. [33]These authors contributed equally: Joan Garcia-Porta, Iker Irisarri, Miguel Vences, Katharina C. Wollenberg Valero. [34]Deceased: Bayram Göçmen.

