## [Peer Review File · Nature Communications]

Reviewers' Comments:

Reviewer #1:

Remarks to the Author:

This paper by Garcia-Porter et al. is a marker-based study to reconstruct the history of Lacertid lizards and correlate it with climate events and molecular evolutionary selection on genes in a limited (700-gene) sample of the transcriptome.

They propose the following four hypotheses (paraphrased):

H1) Increased genetic character sampling will improve estimation of species relationships and divergence times

H2) Extant Lacertidae should exhibit no correlation between climate niche and preferred body temperature due to their active use of sunning.

H3) Decreased rates of speciation and/or increase in extinction during the Cenozoic cooling.

H4) Genes in known vertebrate climate adaptation pathways should show signatures of positive selection, particularly to cold.

Overall, the subject matter is interesting and adds some key points about the relatively understudied temperate groups within squamates. However, this paper's results are mixed and the reporting and discussion extrapolates perhaps too far in many places from a fairly limited data set. A few key conclusions are drawn in the negative and the functional genomic analyses may be suffering from a too small orthologous gene set (~700 genes). The writing could also use one more pass, primarily in several places where longer sentences are overloaded with too many clauses and need to be split or simplified. If the authors were able to focus more on the climatic reconstruction data and hem the reporting closer to the firm quantitative conclusions or try to make stronger functional genomic connections, the manuscript would be greatly enhanced.

I do not recommend this manuscript for publication in its present organization and state, but could see possibly recommending a revised version.

Comments:

1) H1 is not really a hypothesis per se, in that it was not falsifiably tested. Certainly, improving the estimation of the tree was an important goal of the study, but I would not include it in the enumerated "hypotheses." Recommend simply stating it's a goal of the study, and reducing to 3 formal hypotheses.

2) One of the key parts of H2 (and brought up in discussion) was that use of solar heating was the specific mitigating factor in reducing the quantitative relationship between ambient temperature and thermal preference. However, solar radiation data was not analyzed in the ecological correlation data that I could see in the methods or supplement. While certainly a lack of correlation between mean ambient temperature and preferred temperature could point to sunning, this is drawing a conclusion both indirectly and from a lack of evidence in the negative (instead of directly via a significant correlative relationship). This makes this finding stand out as weakly supported by the quantitative data analyzed. Recommend attempting to make this connection more direct (with solar radiation data possibly?) or at least contextualizing the findings more specifically, since this inference on a lack of correlation which could have numerous explanations.

3) H3 seems to be addressed sufficiently in the results.

4) H4 (as stated in the intro) was not addressed by a strong case at all. Mitochondrial stress genes could be responding to a vast array of factors and no single gene involved explicitly in cold response or tolerance was highlighted. This seems very thin evidence to even attempt to answer this question. This seemed more like a broad characterization, where the only quantitative evidence was of some kind of functional effect due to the observed clustering of genes into related genes. However, a null expectation that genes under selection would not be related in some kind of way seems like a straw-man as selection would likely act on gene networks in this way. Again, nothing here specifically pointed to cold, or the connection was not explicitly in the text. Recommend revising the explanation or reclassifying this as a characterization and not a hypothesis.

On ll.256-259, the authors note that the reconstructed ancestral values were more similar for two groups than the individual data values. In such tree models, typically the reconstructed values would be so in any kind of ancestral reconstruction method that largely will cause divergent tip values to converge on interior nodes. Recommend not reporting this as a substantial result, or at least mitigating with a statement noting this property.

- ll.269-270. "This physiological key trait might therefore be phylogenetically conserved especially in the Eremiadini." What does it mean for a trait to be *especially* phylogenetically conserved? Recommend a more precise statement.

- Figure 2 shows correlation plots. Strongly Recommend including fit lines with slope and r^2 values on the plots.

Small notes:

l.101: Missing comma

l.102, 108: consider parentheses for (i.e.,) instead of commas

l. 180: should this be "characterized by signatures"?

l.294: "higher... in recent times" (comparative without two objects. Compared to what?)

l.490-496: run-on sentence, followed by another semicolon-ed run-on.

Reviewer #2:

Remarks to the Author:

Garcia-Porta et al. conduct detailed phylogenetic, biogeographic, and physiological studies of a group of lizards. This study includes impressive new data for these lizards, including a new phylogenomic dataset generated from RNAseq data as well as a new database of distributional data and new physiological data on temperature and water loss tolerances for dozens of species. I think this is a nice example of detailed study of climate evolution in a well-known group of vertebrates. I did not identify any major analytical flaws in the individual analyses involved in generating the phylogeny and associated physiological and comparative analyses, but will note that some of these analyses are likely to be controversial (e.g., estimates of speciation and extinction rate, and inference of patterns of physiological evolution without complete taxon sampling).

Considered together, the data and analyses in this report are used to suggest that this clade of lizards will be unable to tolerate climate change because it adapted its present climatic preferences millions of years ago and will not be able to adapt or change to global warming. Although this is a data-rich and interesting study, I think the conclusion that lacertids are "toast" because their physiological preferences have been relatively constant over the last few million years is well supported by the data. The fact that these lizards may have had relatively stable thermal preferences throughout most of their histories does not mean that they are incapable of surviving climate change through adaptation

via natural selection or other mechanisms. In this context, the brief mention of actual climate change associated declines in lacertids (lines 477-489) provides more convincing evidence for the problems that may be yet to come than the type of speculation based on prior evolutionary history that is focal point of this manuscript. I also think the potentially dire implications of this study do not devote sufficient consideration to the fact that most of the clades and even individual species included in this work appear to have survived far more serious climate change that has occurred over the past 20 million years than is likely to result from human and resulting global warming.

For the reasons noted above, I do not believe that this manuscript provides strong evidence to support its provocative conclusion that Palearctic lizards are "toast." In particular, I do not believe the evidence presented here is strong enough to result in a contribution that will be of extreme importance to scientists in the field of global climate change.

In addition to not being entirely convinced by this papers main conclusions, I think this paper devotes to much attention to issues that are unlikely to be of widespread interest to readers of Nature, such as reconstruction of a well-resolved tree for lacertid lizards. I would also have liked to have seen more sophisticated analyses that rates of evolution differ across climatic conditions, including some discussion of whether those genes that are specifically expected to be involved in climatic adaptation show particularly noteworthy patterns of evolution relative to genes that are not expected to be involved in climatic adaptation.

Manuscript: NCOMMS-18-27080-T (Garcia-Porta et al.)

New title: Thermal physiology, diversification and genome-wide substitution rates shaped by past and present environmental temperatures in a clade of heliothermic lizards

Author responses to reviewer comments

We are grateful to the reviewers for their constructive comments, and to the editors for the opportunity to resubmit an entirely revised version of this manuscript for consideration.

Both reviewers raised a number of issues with our initial manuscript, and this led us to reconsider some of our main conclusion. The main point of both reviewers was that our main focus was insufficiently reflected in our analyses, and that therefore, our main conclusions were insufficiently supported by our data. Their arguments were convincing and led us to agree with their opinion. Consequently, we have now re-focused the manuscript, and our conclusions are now based on those results that were most strongly supported, and to exclude some analyses (like the functional genetics/selection analysis) which were less clear. We also revised the list of references, adding a substantial body of new literature published in 2018 and 2019, and emphasized more clearly why lacertids are an important and highly suitable group to study the effects of past and present climatic conditions on lizard evolution.

Furthermore, we carried out a number of major (re-)analyses which might not be visible at first but which took a considerable effort: (1) we included bioclimatic variables related to solar radiation, as suggested by one reviewer, and repeated the respective statistical analyses based on the new data set; (2) we added a summary of field body temperatures to strengthen our case that the measured T_{pref} values are biologically meaningful; (3) we newly computed the spatial analyses with a novel, more suitable dataset of the GARD initiative which became available at the end of 2018; (4) we added analyses of range sizes based on the GARD data; and (5) we computed root-to-tip paths (indicative of substitution rates) for the 5878 single-gene trees of the phylotranscriptomic data and find evidence for a genome-wide deceleration of substitution rates in cold-adapted species.

Despite these new analyses, we also made an effort to make the manuscript more concise, and it is now substantially shorter than previously. Depending on the recommendations of reviewers and editors, we would of course be able to further condense the manuscript, if it was deemed suitable for Nature Communications.

In the following, we provide specific responses to the original reviewer comments.

Reviewer #1 (Remarks to the Author):

This paper by Garcia-Porter et al. is a marker-based study to reconstruct the history of Lacertid lizards and correlate it with climate events and molecular evolutionary selection on genes in a limited (700-gene) sample of the transcriptome.

They propose the following four hypotheses (paraphrased):

H1) Increased genetic character sampling will improve estimation of species relationships and divergence times

H2) Extant Lacertidae should exhibit no correlation between climate niche and preferred body temperature due to their active use of sunning.

H3) Decreased rates of speciation and/or increase in extinction during the Cenozoic cooling.

H4) Genes in known vertebrate climate adaptation pathways should show signatures of positive selection, particularly to cold.

Overall, the subject matter is interesting and adds some key points about the relatively understudied temperate groups within squamates. However, this paper's results are mixed and the reporting and discussion extrapolates perhaps too far in many places from a fairly limited data set. A few key conclusions are drawn in the negative and the functional genomic analyses may be suffering from a too small orthologous gene set (~700 genes). The writing could also use one more pass, primarily in several places where longer sentences are overloaded with too many clauses and need to be split or simplified. If the authors were able to focus more on the climatic reconstruction data and hem the reporting closer to the firm quantitative conclusions or try to make stronger functional genomic connections, the manuscript would be greatly enhanced.

I do not recommend this manuscript for publication in its present organization and state, but could see possibly recommending a revised version.

Response: Thank you for these constructive general comments, and for your willingness to consider a revised manuscript version. As said above, we have given this substantial thought and accept your concerns. The new manuscript has been substantially revised and rewritten, and we here would like to mention the two main points: we decided to drop the functional genomic analyses as we found it difficult to improve this part, and instead have focused more strongly on those aspects of the climatic reconstruction that we consider to be well supported. We have also tried to improve the writing, but we are aware that this part might require additional attention if we were given the opportunity of a second revision.

Comments:

1) H1 is not really a hypothesis per se, in that it was not falsifiably tested. Certainly, improving the estimation of the tree was an important goal of the study, but I would not include it in the enumerated "hypotheses." Recommend simply stating it's a goal of the study, and reducing to 3 formal hypotheses.

Response: We fully agree. We have now removed the formal listing of hypotheses and instead state the goals of the study in a more generalized paragraph.

2) One of the key parts of H2 (and brought up in discussion) was that use of solar heating was the specific mitigating factor in reducing the quantitative relationship between ambient temperature and thermal preference. However, solar radiation data was not analyzed in the ecological correlation data that I could see in the methods or supplement. While certainly a lack of correlation between mean ambient temperature and preferred temperature could point to sunning, this is drawing a conclusion both indirectly and from a lack of evidence in the negative (instead of directly via a significant correlative relationship). This makes this finding stand out as weakly supported by the quantitative data analyzed. Recommend attempting to make this connection more direct (with solar radiation data possibly?) or at least contextualizing the findings more specifically, since this inference on a lack of correlation which could have numerous explanations.

Response: This is certainly a very good suggestion for which we are grateful. We have now included solar radiation layers and repeated a great part of our analyses for which these new data were relevant. These new analyses revealed that (1) lacertid species richness is geographically strongly predicted by solar radiation, clearly defining these lizards as heliotherms; (2) however for most correlations with physiological, genomic and range size patterns, temperature appeared to be a more important factor than solar radiation, pointing to the fact that these lizards are more strongly affected by general environmental temperature than would be expected if they would represent full thermoregulators.

3) H3 seems to be addressed sufficiently in the results.

Response: Thank you for this comment. In any case, we have also made an attempt to better integrate these results with those of other analyses.

4) H4 (as stated in the intro) was not addressed by a strong case at all. Mitochondrial stress genes could be responding to a vast array of factors and no single gene involved explicitly in cold response or tolerance was highlighted. This seems very thin evidence to even attempt to answer this question. This seemed more like a broad characterization, where the only quantitative evidence was of some kind of functional effect due to the observed clustering of genes into related genes. However, a null expectation that genes under selection would not be related in some kind of way seems like a straw-man as selection would likely act on gene networks in this way. Again, nothing here specifically pointed to cold, or the connection was not explicitly in the text. Recommend revising the explanation or reclassifying this as a characterization and not a hypothesis.

Response: Good point. We revisited the functional genomic data set and we agree that at present and in the context of the present paper, this data set adds little to the overall picture and does not provide any convincing evidence for an adaptation to cold.

On ll.256-259, the authors note that the reconstructed ancestral values were more similar for two groups than the individual data values. In such tree models, typically the reconstructed values would be so in any kind of ancestral reconstruction method that largely will cause divergent tip values to converge on interior nodes. Recommend not reporting this as a substantial result, or at least mitigating with a statement noting this property.

Response: Good point. We fully agree and have dropped this statement altogether.

- ll.269-270. "This physiological key trait might therefore be phylogenetically conserved especially in the Eremiadini." What does it mean for a trait to be *especially* phylogenetically conserved? Recommend a more precise statement.

Response: We agree this statement was not precise. We have opted to delete this sentence in a general effort to condense the manuscript.

- Figure 2 shows correlation plots. Strongly Recommend including fit lines with slope and r^2 values on the plots.

Response: This is a very valid point. We have added the fit lines as requested. Furthermore, we have exchanged one of the plots with a new one that illustrates the genome-wide trend in temperature-dependent substitution rates.

Small notes:

l.101: Missing comma

Response: This sentence has been rephrased, paying attention to avoid missing commas.

l.102, 108: consider parentheses for (i.e.,) instead of commas

Response: Thank you for this recommendation. We have modified this throughout as suggested.

l. 180: should this be "characterized by signatures"?

Response: This sentence has been rephrased.

l.294: "higher... in recent times" (comparative without two objects. Compared to what?)

Response: Rephrased to " an increase of climatic disparity in recent times"

l.490-496: run-on sentence, followed by another semicolon-ed run-on.

Response: This whole section has been completely rephrased.

Reviewer #2 (Remarks to the Author):

Garcia-Porta et al. conduct detailed phylogenetic, biogeographic, and physiological studies of a group of lizards. This study includes impressive new data for these lizards, including a new phylogenomic dataset generated from RNAseq data as well as a new database of distributional data and new physiological data on temperature and water loss tolerances for dozens of species. I think this is a nice example of detailed study of climate evolution in a well-known group of vertebrates. I did not identify any major analytical flaws in the individual analyses involved in generating the phylogeny and associated physiological and comparative analyses, but will note that some of these analyses are likely to be controversial (e.g., estimates of speciation and extinction rate, and inference of patterns of physiological evolution without complete taxon sampling).

Response: Thank you for this overall positive assessment of the data. We are aware that some of the analyses might be controversial, but in the new manuscript version we have made an effort to concentrate on those results that are most strongly supported. If given the opportunity for a second revision, we will be glad to answer to more specific queries about the analytical procedures and provide arguments why we consider them to be sufficiently robust.

Considered together, the data and analyses in this report are used to suggest that this clade of lizards will be unable to tolerate climate change because it adapted its present climatic preferences millions of years ago and will not be able to adapt or change to global warming. Although this is a data-rich and interesting study, I think the conclusion that lacertids are “toast” because their physiological preferences have been relatively constant over the last few million years is well supported by the data. The fact that these lizards may have had relatively stable thermal preferences throughout most of their histories does not mean that they are incapable of surviving climate change through adaptation via natural selection or other mechanisms. In this context, the brief mention of actual climate change associated declines in lacertids (lines 477-489) provides more convincing evidence for the problems that may be yet to come than the type of speculation based on prior evolutionary history that is focal point of this manuscript. I also think the potentially dire implications of this study do not devote sufficient consideration to the fact that most of the clades and even individual species included in this work appear to have survived far more serious climate change that has occurred over the past 20 million years than is likely to result from human and resulting global warming.

Response: These are important and very valid points. We have addressed these concerns in multiple ways. Most importantly we have (1) completely re-focused the manuscript and de-emphasized the "lizards are toast" conclusions, (2) instead provided more specific arguments why we think our data confirm that a particular sub-group (microendemic, montane taxa) are most threatened among this group of lizards, and based this conclusion (3) on new data and analyses on range sizes and climatic disparity evolution (more clearly emphasizing that the majority of these lizards evolved their adaptations in a period of global cooling, not stability).

For the reasons noted above, I do not believe that this manuscript provides strong evidence to support its provocative conclusion that Palearctic lizards are “toast.” In particular, I do not believe the evidence presented here is strong enough to result in a contribution that will be of extreme importance to scientists in the field of global climate change.

Response: We agree with the first statement and have strongly de-emphasized this conclusion. While it might be exaggerated to say that our study is of "extreme importance" we nevertheless think that it provides very important baseline data to understand the origin of physiological adaptations in lizards and to assess and predict the biotic effects of rising or falling global temperatures.

In addition to not being entirely convinced by this paper's main conclusions, I think this paper devotes too much attention to issues that are unlikely to be of widespread interest to readers of *Nature*, such as reconstruction of a well-resolved tree for lacertid lizards. I would also have liked to have seen more sophisticated analyses that rates of evolution differ across climatic conditions, including some discussion of whether those genes that are specifically expected to be involved in climatic adaptation show particularly noteworthy patterns of evolution relative to genes that are not expected to be involved in climatic adaptation.

Response: Thank you for these comments – this quite strong criticism pointed us to several weak points in our manuscript, and thereby led us to improve it in several ways.

First, we have made an effort to strengthen our case in the introduction why lacertids, and clarifying their phylogenetic relationships, is of importance for the community. In addition to this, we might here mention that a keyword search in "Scopus" (combined for "Lacertidae", "lacertid" and a series of important genera) yielded 4140 hits, whereas a search for "Anolis" or "anole" yielded 2335 hits. Obviously, this does not demonstrate lacertids to be a more important or more suitable lizard model than anoles which are unparalleled in their adaptive radiation and convergent adaptation to ecomorphospaces, thus providing fantastic insights into evolutionary processes. However, the large number of scientific publications dealing with lacertids highlight these lizards as an intensively studied group, of interest to a large community of scientists --- and for many of these, a reliable phylogeny is paramount to interpret their results.

Secondly, also for the molecular evolution section, we decided to drop those parts that we considered provided only weak evidence (i.e., genes under selection). Instead performed a more detailed analysis of the deceleration of molecular substitution rates related to cold environments, because this part of the data revealed a particularly robust pattern. We analyzed for all 5878 single-gene trees the root-to-tip paths in relation to temperature and found that this relation applies to the vast majority of them, suggesting that the effect of temperature on molecular substitution rates is genome-wide (new Fig. 2E). To our knowledge, this is one of the most comprehensive data sets on which this pattern has been analyzed so far, and among those revealing the best supported pattern. While an impact of this pattern on the potential for fast climate change adaptation remains to be demonstrated (and indeed may be doubted), it does provide a striking example for the impact of environmental temperature on the evolution of genomic traits.

Reviewers' Comments:

Reviewer #1:

Remarks to the Author:

Manuscript is much improved from the first draft, both in writing flow and scientific pitch.

The evidence presented is much more solid in supporting the discussion points, and I think the dropping of the "toast" aspect is positive and gives the manuscript a more neutral but convincing tone.

Without the molecular connection, the paper is now highly dependent on climate regression and speciation rate imputation models (both of which have huge error bars), which means (as the other reviewer noted) this paper will likely be controversial.

However, the analyses are thorough and conducted with effective methods, and the results are now stated much more carefully. This expands the view of climatic effects to squamates in a clear and interesting way.

I can find no further scientific or stylistic suggestions, and recommend this paper for publication.

Reviewer #3:

Remarks to the Author:

The authors test for associations between climatic data and physiology and molecular evolution using Lacertid lizards. This included using a large molecular data set to first establish relationships among species in this group. The phylogenetic hypotheses they construct are largely in accordance with current grouping, bolstering our understanding of the relationships within this clade. They found species richness to be low in aseasonal environments, and highest in areas with the greatest solar radiation. Rates of molecular evolution were estimated to be faster in groups that occupy warmer conditions, consistent with predictions from the Metabolic Theory of Ecology. Species from warmer environments were also found to have larger ranges. Thermal physiology (preferred body temperature and water loss rates) correlated with climatic conditions, and in particular preferred temperatures were positively correlated with broad-scale temperature. The authors interpret the latter result to suggest that many of the species are thermoconformers that will be vulnerable to changes conditions, particularly those living in cool habitats.

Overall I found this to be an interesting manuscript with a nice integration of phylogenetic, climatic and physiological data. Many of the results have important implications for the effects of habitat temperature on evolutionary processes. There are some areas where I was confused by the predictions, and where the inferences didn't match the results, particularly with respect to the evolution of thermoconformity and the threat of global change to this group. Specific comments are below.

Lines 173-178 Confused here. Say you expect evolution to be primarily driven by adaptation to cold, but then predict lower diversification rates during cold periods. Wouldn't you expect greater diversification if cold was driving divergence?

Line 238 – I don't see how you are testing anything about thermoconformity with bioclimatic niche data.

Line 371 – Tpref correlating with bioclimatic conditions is not evidence of thermoconformity. It may be

evidence that thermoregulation can't maintain constant body temperatures across different habitats, but that does not mean the animals are not working hard to thermoregulate in all of those different habitats. Species in cold environments often bask the most but can still experience cooler average body temperatures.

Lines 378 – 380 The environment/ T_{pref} correlation is not a surprising result. Behavioral thermoregulation can slow evolutionary change in thermal physiology, but published data across lizards and many other ectotherms make clear that it doesn't prevent it all together.

Line 455 – With respect to physiology, your data show correlations between current physiological traits and current environmental temperatures only. There are no explicit tests of associations between past physiological traits and past climatic conditions.

Line 460 – See comment about thermoconformity above.

Line 460-462 Not following the inference that cool adapted species are at risk from warming based on your data. T_{pref} was evolving while things were cooling. So? Are these species living in environments closer to their thermal limits than others? That was not tested. Maybe it will help them, as they won't have to expend as much time/energy basking when air temperatures are low.

Line 556 – Need to see explicit methods for how IWL was corrected for body area.

Reviewer #1 (Remarks to the Author):

Manuscript is much improved from the first draft, both in writing flow and scientific pitch.

The evidence presented is much more solid in supporting the discussion points, and I think the dropping of the "toast" aspect is positive and gives the manuscript a more neutral but convincing tone.

Without the molecular connection, the paper is now highly dependent on climate regression and speciation rate imputation models (both of which have huge error bars), which means (as the other reviewer noted) this paper will likely be controversial.

However, the analyses are thorough and conducted with effective methods, and the results are now stated much more carefully. This expands the view of climatic effects to squamates in a clear and interesting way.

I can find no further scientific or stylistic suggestions, and recommend this paper for publication.

Response: Thank you for this positive evaluation – we are glad the reviewer agrees the new manuscript is more solid in its conclusions. Following the remark of the reviewer, we highlighted in the Discussion more clearly the uncertainty of some of our inferences, especially due to the

error inherent in molecular clock analyses. We also added information on the error bars in the speciation plot to the figure legend as we noted this information was missing.

Reviewer #3 (Remarks to the Author):

The authors test for associations between climatic data and physiology and molecular evolution using Lacertid lizards. This included using a large molecular data set to first establish relationships among species in this group. The phylogenetic hypotheses they construct are largely in accordance with current grouping, bolstering our understanding of the relationships within this clade. They found species richness to be low in aseasonal environments, and highest in areas with the greatest solar radiation. Rates of molecular evolution were estimated to be faster in groups that occupy warmer conditions, consistent with predictions from the Metabolic Theory of Ecology. Species from warmer environments were also found to have larger ranges. Thermal physiology (preferred body temperature and water loss rates) correlated with climatic conditions, and in particular preferred temperatures were positively correlated with broad-scale temperature. The authors interpret the latter result to suggest that many of the species are thermoconformers that will be vulnerable to changes conditions, particularly those living in cool habitats. Overall I found this to be a an interesting manuscript with a nice integration of phylogenetic, climatic and physiological data. Many of the results have important implications for the effects of habitat temperature on evolutionary processes. There are some areas where I was confused by the predictions, and where the inferences didn't match the results, particularly with respect to the evolution of thermoconformity and the threat of global change to this group. Specific comments are below.

Response: Thank you for this overall positive evaluation. We have tried to clarify and partly modified predictions and inferences. In particular we have removed our initial references to thermoconformity which we agree were not accurate, and have added additional analyses (new Fig. 4) to address both the thermoregulator/thermoconformer issue and the threat of global change more tightly to our data. See below for more details on these points.

Lines 173-178 Confused here. Say you expect evolution to be primarily driven by adaptation to cold, but then predict lower diversification rates during cold periods. Wouldn't you expect greater diversification if cold was driving divergence?

Response: This is a good point. We are grateful to the reviewer for raising this question which led us to rephrase our diversification / adaptation hypothesis more precisely and with more accurate wording.

There are two main issues that we believe caused the concern of the reviewer:

The first is the distinction between speciation on one hand, and adaptation (to climate) on the other hand. Both are evolutionary processes, and using the term "evolution" lacks precision. We have rephrased our hypotheses in the introduction to clearly separate these two processes.

The second issue is which of these processes we expect to be driven by cold, and which would be the underlying mechanisms. We here lay out our arguments in some detail:

- Certainly we can expect adaptation to cold: Ancestral lacertids were adapted to warm conditions during the Paleocene/Eocene, and then were forced to cope with cooler climates in the subsequent epochs which they likely achieved with a whole array of behavioral, morphological and physiological adaptations. This adaptive process can be expected to be triggered by drastic cooling events when selection was strongest, but could also occur in times of stable cool conditions (e.g., through warm-adapted lizards expanding their range into cool regions).

- To formulate a hypothesis on climate-driven diversification, it first is necessary to obtain insights into speciation modes in these lizards. We calculated range overlap values derived from the GARD data set, and literature data, to estimate allopatric vs. sympatric occurrence of 68 pairs of lacertid sister species for which reliable data were available. The majority of these (42) were allopatric, and an additional 12 were sympatric only in relatively small proportions of their range, of 10% of the added range of both species. This suggests that speciation of lacertids takes place in many, if not most cases under allopatric conditions. Allopatry can be driven by rapid climatic change, that is, by restricting cold-adapted species to mountain refugia during warming (or restricting warm-adapted species to lowland refugia separated by mountains). Therefore, we would expect lacertid diversification mainly in periods of either global cooling or warming – and because increases in speciation rate were mainly in the cold-adapted Lacertini, which contain many montane microendemics, it makes sense to expect diversification being triggered by warming.

To accommodate these points, we have reworded our initial hypothesis because we understand that our initial text was prone to misunderstanding. Rather than a slowdown of diversification with cooling we now refer to an increase of diversification with warming. In addition, we have also modified the results to emphasize the possible coincidence of the rise in Lacertini species to a warming episode in the mid-Miocene.

Line 238 – I don't see how you are testing anything about thermoconformity with bioclimatic niche data.

Response: We agree and have removed this statement.

Line 371 – T_{pref} correlating with bioclimatic conditions is not evidence of thermoconformity. It may be evidence that thermoregulation can't maintain constant body temperatures across different habitats, but that does not mean the animals are not working hard to thermoregulate in all of those different habitats. Species in cold environments often bask the most but can still experience cooler average body temperatures.

Response: We agree our use and definition of "thermoconformers" in the previous manuscript version was inaccurate. To explain how the confusion arose, the evolutionary biologists in the author team interpreted this term in a more evolutionary sense - i.e., "evolutionary thermoconformers" would be species that adapt their T_{pref} to the environmental conditions they experience, in an evolutionary timescale. However, it is obvious that this definition is not the

prevailing one and is not shared by physiologists and ecologists, and we have therefore completely dropped it.

Instead, we have added an additional analysis that addresses at the same time the question of thermoconformity and threats from climate warming (see response further below).

We use a simple metric to compare the degree of thermoconformity versus heliothermy among lacertids. As thermal preference (T_{pref}) converges on ambient daily maximum air temperature, a typical heliothermic lizard can assume a more thermoconforming behavior. This transition typically occurs at lower latitudes and altitudes as observed in *Anolis* and *Tropidurus* lizards (Huey 1982, Huey and Webster 1976, Piantoni et al. 2016) more typical of forested environments with limited opportunity for behavioral thermoregulation. Therefore, the difference between thermal preference and daily maximum air temperature, T_{Max} , averaged across months, is a key condition of thermoconformity. However, in temperate and boreal sites, lizards retreat in winter, so we constrain this calculation to months where daily $T_{min} > 4C$ (T_{min} and T_{Max} , from worldclim.org), so there is a limited risk of freezing in nighttime retreats. At more equatorial sites (i.e., either the dry-wet tropics of the equatorial tropics, degrees latitude > 25), thermoregulating by trying to find gaps of lights in the forest is far costlier than adopting a thermoconforming strategy (Huey and Slatkin 1976). In the figure below, we compute this metric of thermoconformity (values close to zero) vs. heliothermy (values further from zero) among lacertids (29919 occurrence records for 54 species with T_{pref} values).

The plot shows that lacertid lizards outside the equatorial tropics, do have larger differences between their preferred or body temperature and the environmental temperature, which means they must be heliotherms. While on the other hand, numerous lizard populations in the tropics have T_{pref} close to T_{max} and thus can be expected to be partial thermoconformers, i.e., they need to thermoregulate only during a lower amount of time.

We have added one new figure (Fig. 4) and a short section reporting this analysis. We also have modified throughout the manuscript all sentences referring to thermoconforming/thermoregulating behavior.

References

- Huey, R.B. (1982). Temperature, physiology, and the ecology of reptiles. In *Biology of the Reptilia*, volume 12, physiology C: 25–91. Gans, C. & Pough, F.H. (Eds). London: Academic Press.
- Huey, R.B. and Slatkin, M., 1976. Cost and benefits of lizard thermoregulation. *The Quarterly Review of Biology*, 51(3), pp.363-384.
- Huey, R.B. & Webster, T.P. (1976). Thermal biology of *Anolis* lizards in a complex fauna: the *crisatellus* group on Puerto Rico. *Ecology* 57, 985–994.#
- Piantoni, C., Navas, C.A. and Ibarquengoytía, N.R., 2016. Vulnerability to climate warming of four genera of New World iguanians based on their thermal ecology. *Animal Conservation*, 19(4), pp.391-400.
- Campbell-Staton, S.C., Cheviron, Z.A., Rochette, N., Catchen, J., Losos, J.B. and Edwards, S.V., 2017. Winter storms drive rapid phenotypic, regulatory, and genomic shifts in the green anole lizard. *Science*, 357(6350), pp.495-498.

Lines 378 – 380 The environment/Tpref correlation is not a surprising result. Behavioral thermoregulation can slow evolutionary change in thermal physiology, but published data across lizards and many other ectotherms make clear that it doesn't prevent it all together.

Response: The reviewer is right that the pattern itself - physiological traits differing along a climatic gradient - is not surprising, and yes, there are published data for lizards that indicate such a trend. However, these analyses usually are studies based on multiple populations of a single species, or a meta-analysis across all lizards. Our study provides conclusive evidence from a multi-species analysis across one lizard family occurring in different climatic regions - to our knowledge, it is the only such comprehensive analysis to date (although we refrained from emphasizing this in the manuscript). Furthermore, we still find it striking that for many traits, it is indeed the correlation with temperature - and not, for instance, solar radiation - and that the correlation is so consistent for some traits such as molecular substitution rates.

We have reworded the text so that we explicitly state that the correlation between climate and physiology is an expected pattern, but that the consistency of the encountered correlations makes these results worth reporting.

Line 455 – With respect to physiology, your data show correlations between current physiological traits and current environmental temperatures only. There are no explicit tests of associations between past physiological traits and past climatic conditions.

Response: Testing for correlations between past traits and climatic conditions is tricky because of the added uncertainties that would plague correlations between reconstructed ancestral traits and distribution ranges of past lineages (because the reconstructed ancestral states depend strongly on the states of extant taxa). Furthermore such analyses cannot be readily corrected for phylogeny. We did the respective (not phylogenetically corrected) analyses which show a clear (and significant) correlation (see scatterplot below), but since the underlying data are so strongly dependent from the extant data, we prefer not including this analysis in the manuscript.

To address the reviewer concern, we removed "past" from the respective line, and have also specified elsewhere that the correlations performed referred only to values of extant species.

To address the reviewer concern on this specific topic, we removed "past" from the respective line, and have also specified elsewhere that the correlations performed referred only to values of extant species.

Next, however, the reviewer comment made us wonder if there is a way to indeed document the evolutionary trajectory of physiological and niche adaptations over time. These trajectories already in the original manuscript were graphically represented in the colored circle trees (Fig. 2A and S12, but it is indeed difficult to "intuitively" extract an overall pattern from these trees.

We therefore computed the value for Tpref and hours>30°C for each branch as the average between the two nodes connecting to it, and then averaged the values for all branches (lineages) existing at a certain point in time. We did not do this analysis for IWL due to the unbalanced representation of Eremiadini vs. Lacertini in this data set. The results, now added to Fig. 2B, show clearly a decrease of niche temperature and Tpref in lacertids after 30 Ma, mostly due to the rise of the Lacertini, with the origin of more and more lineages adapted to cool temperatures. It is rather striking how this pattern fits the Cenozoic trend of global cooling temperatures represented in Fig. 2B – although the respective data points were calculated completely independently from the paleoenvironmental data. These new results therefore reinforce our inference of ancient physiological and niche adaptations to cold in the Lacertidae.

Line 460 – See comment about thermoconformity above.

Response: See our response above. We agree our use of "thermoconformity" was inaccurate and prone to misunderstanding, and the text has been modified accordingly.

Line 460-462 Not following the inference that cool adapted species are at risk from warming based on your data. Tpref was evolving while things were cooling. So? Are these species living in environments closer to their thermal limits than others? That was not tested. Maybe it will help them, as they won't have to expend as much time/energy basking when air temperatures are low.

Response: We addressed this comment with the same analytical approach as the thermoconformity issue (see above).

The putative selective advantage of lowering T_{pref} relative to T_{max} would allow lizards to spend less time engaged in thermoregulatory behavior in cooler environments. If environmental temperatures rise, however, the consequence could be a lower thermal safety margin. But at the same time, a large number of occurrence records below the 30° latitudes fall very close to the environmental temperature, which translates into a smaller thermal safety margin, also alluded to by the reviewer but not explicitly stated in our manuscript. These taxa are perilously close to extinction under future climate scenarios, as T_{max} would exceed T_{pref} . In the temperate zone, T_{pref} is much more distant from T_{max} and thus, temperate lacertids in principle have a greater thermal safety margin. Still, numerous local extinctions have been reported from temperate lacertids.

We therefore also explored the utility of the T_b/T_{max} environmental metric in predicting contemporary extinctions that have already been mapped for the cold-adapted *Zootoca vivipara* (Sinervo et al. 2010). We find the predicted metric well predicts the observed extinctions of *Z. vivipara* (red points in the graph below) relative to the other occurrence records (blue points), despite at some of these locations the thermal safety margin was still rather high ($>10^{\circ}\text{C}$). These extinctions may be triggered not only by temperature but also by increased aridity as *Z. vivipara* is a species with high IWL and additionally, viviparous in many of its populations.

We have added the new Fig 4 and a short paragraph in Results, and revised the Discussion throughout to include these aspects.

Line 556 – Need to see explicit methods for how IWL was corrected for body area.

Response: This information was previously included in the supplementary materials, but we followed the reviewer's suggestion and moved the formula (plus the respective reference by Grigg et al.) to the main manuscript.

Reviewers' Comments:

Reviewer #3:

Remarks to the Author:

I appreciate the effort the authors have put into revising this manuscript. As stated in my first review, I find this to be an interesting manuscript with a nice integration of phylogenetic, climatic and physiological data. More specific comments below.

Line 381 – State the direction of the associations to communicate how they support your specific hypotheses.

Line 388 – If the association is in the direction you predict, how can it be surprising? What effect size would you expect, and how does your effect size differ from it?

Line 467 – As with the first version, the discussion of climate change risk is by far the weakest link in this manuscript. Given how extensive the other analyses are and the many other interesting results that the authors found, why end on a discussion of something that you simply do not have the data to test? Fig 1B is sort of interesting, but the threshold for extinction indicated is still a large safety margin. And so many other populations, even at high latitude, are below that threshold, and sometimes by a lot (Fig 1a), and no evidence is presented that those populations are doing particularly poorly. This overall lack of evidence leads to a lot of hand-waving about interactions between evaporative water loss and water availability and thermoregulation that are simply not supported. I recommend the authors remove this part of the paper, including Fig. 4, and focus on the strong results that they have.

Figure legends 2 and 3 begin with "The influence of..." However, these are correlative analyses that do not show causation. They should say something like "Associations between..."

Manuscript NCOMMS-18-27080B

Title: "Environmental temperatures shape thermal physiology, diversification and genome-wide substitution rates in lizards"

Author's responses to reviewer comments and editorial requests

Response: We appreciate the effort of the reviewers in evaluating this manuscript, and of the editor to provide detailed suggestions of how to make the manuscript suitable for the standards of "Nature Communication".

REVIEWERS' COMMENTS:

Reviewer #3 (Remarks to the Author):

I appreciate the effort the authors have put into revising this manuscript. As stated in my first review, I find this to be an interesting manuscript with a nice integration of phylogenetic, climatic and physiological data. More specific comments below.

Line 381 – State the direction of the associations to communicate how they support your specific hypotheses.

Reply: Modified as suggested.

Line 388 – If the association is in the direction you predict, how can it be surprising? What effect size would you expect, and how does your effect size differ from it?

Reply: We have rephrased this paragraph again and emphasize more clearly now that the overall pattern was NOT surprising as it agreed with our hypotheses, but that the consistency by which temperature associations with very different traits showed up was striking.

Line 467 – As with the first version, the discussion of climate change risk is by far the weakest link in this manuscript. Given how extensive the other analyses are and the many other interesting results that the authors found, why end on a discussion of something that you simply do not have the data to test? Fig 1B is sort of interesting, but the threshold for extinction indicated is still a large safety margin. And so many other populations, even at high latitude, are below that threshold, and sometimes by a lot (Fig 1a), and no evidence is presented that those populations are doing particularly poorly. This overall lack of evidence leads to a lot of hand-waving about interactions between evaporative water loss and water availability and thermoregulation that are simply not supported. I recommend the authors remove this part of the paper, including Fig. 4, and focus on the strong results that they have.

Reply: We agree with this main concern and did not want to give the impression to make unwarranted statements. As suggested by the editor, we have moved the new Fig. 4 to the Supplements. We have also rephrased a few sentences (in particular in the second last paragraph of the discussion) to make it clearer that here we are not reporting facts but proposing hypotheses in order to stimulate and direct future work.

Figure legends 2 and 3 begin with “The influence of...” However, these are correlative analyses that do not show causation. They should say something like “Associations between...”

Reply: Modified as suggested.